# Memory Efficient Neural Processes via Constant Memory Attention Block

## Abstract

Neural Processes (NPs) are popular meta-learning methods for efficiently modelling predictive uncertainty. Recent state-of-the-art methods, however, leverage expensive attention mechanisms, limiting their applications, particularly in low-resource settings. In this work, we propose Constant Memory Attention Block (CMAB), a novel general-purpose attention block that (1) is permutation invariant, (2) computes its output in constant memory, and (3) performs updates in constant computation. Building on CMAB, we propose Constant Memory Attentive Neural Processes (CMANPs), an NP variant which only requires **constant** memory. Empirically, we show CMANPs achieve state-of-the-art results on popular NP benchmarks (meta-regression and image completion) while being significantly more memory efficient than prior methods.

## 1 Introduction

Memory efficiency is important for a variety of reasons, for example: (1) Modern hardware, such as GPUs and TPUs, are often memory-constrained for applications and computing attention mechanisms is memory-intensive. This issue is accentuated now due to the ubiquity of low-memory/compute domains (e.g., IoT devices). (2) Memory efficiency is important in embedded platforms where memory access energy intensive. This is particularly important in mobile robots where a limited energy supply needs to be allocated (Li et al., 2022a).

Neural Processes (NPs) have been popular meta-learning methods for efficiently modelling predictive uncertainty. They have been applied to a wide variety of settings such as graph link prediction (Liang & Gao, 2022), continual learning (Requeima et al., 2019), and geographical precipitation modeling (Foong et al., 2020) – many of which can have high-dimensional inputs. NPs are particularly useful in low-resource settings due to not requiring retraining from scratch given new data. State-of-the-art methods, however, rely on attention mechanisms which require a substantial amount of memory and do not scale well with the number of tokens (Jha et al., 2022), limiting their applications in low compute domains (e.g., IoT devices, mobile phones and other battery-powered devices). For example, Transformer Neural Processes (TNPs) (Nguyen & Grover, 2022) scale quadratically with the size of the context and query dataset. Latent Bottlenecked Attentive Neural Processes (LBANPs) (Feng et al., 2023) is $\mathcal{O}(NL)$ where $N$ is the size of the context dataset and $L$ is a hyperparameter that scales with the difficulty of the task and the size of the context dataset.

As such, in this work, we propose (1) Constant Memory Attention Block (CMAB), a novel general-purpose attention block that (i) is permutation invariant, (ii) computes its output in constant memory, and (iii) performs updates in constant computation. To the best of our knowledge, we are the first to propose an attention mechanism with the aforementioned properties. Due to having memory usage independent of the number of inputs, CMABs naturally scale to large amounts of inputs.

Building on CMABs, we propose (2) Constant Memory Attentive Neural Processes (CMANPs). By leveraging the efficiency properties of CMABs, CMANPs are (i) scalable in the number of datapoints and (ii) allow for efficient updates. Leveraging the efficient updates property, we further introduce an Autoregressive Not-Diagonal extension, namely, CMANP-AND which only requires constant memory unlike the quadratic memory required by all prior Not-Diagonal extensions. In the experiments, CMANPs achieve state-of-the-art results on meta-regression and image completion tasks. We empirically show that CMANPs only require constant memory, making it significantly more efficient than prior state-of-the-art methods.

## 2 BACKGROUND

### 2.1 META-LEARNING FOR PREDICTIVE UNCERTAINTY ESTIMATION

In meta-learning for predictive uncertainty estimation, models are trained on a distribution of tasks $\Omega(\mathcal{T})$ to model a probabilistic predictive distribution. A task $\mathcal{T}$ is a tuple $(\mathcal{X}, \mathcal{Y}, \mathcal{L}, q)$ where $\mathcal{X}, \mathcal{Y}$ are the input and output space respectively, $\mathcal{L}$ is the task-specific loss function, and $q(x, y)$ is the task-specific distribution over data points. During each meta-training iteration, $B$ tasks $\mathbf{T} = \{\mathcal{T}_i\}_{i=1}^B$ are sampled from the task distribution $\Omega(\mathcal{T})$. For each task $\mathcal{T}_i \in \mathbf{T}$, a context dataset $\mathcal{D}_C^i = \{(x, y)^{i,j}\}_{j=1}^N$ and a target dataset $\mathcal{D}_T^i = \{(x, y)^{i,j}\}_{j=1}^M$ are sampled from the task-specific data point distribution $q_{\mathcal{T}_i}$. $N$ is a fixed number of context datapoints and $M$ is a fixed number of target datapoints. The model is adapted using the context dataset. Afterwards, the target dataset is used to evaluate the effectiveness of the adaptation and adjust the adaptation rule accordingly.

### 2.2 NEURAL PROCESSES

Neural Processes (NPs) are meta-learned models that define a family of conditional distributions. Specifically, NPs condition on an arbitrary amount of context datapoints (labelled datapoints) and make predictions for a batch of target datapoints, while preserving invariance in the ordering of the context dataset. NPs consist of three phases: conditioning, querying, and updating

**Conditioning:** In the conditioning phase, the model encodes the context dataset $\mathcal{D}_C$. Neural Processes (Garnelo et al., 2018b) model functional uncertainty by encoding the context dataset as a Gaussian latent variable: $z_C \sim q(z|\mathcal{D}_C)$ where $q(z|\mathcal{D}_C) = \mathcal{N}(z; \mu_C, \sigma_C^2)$ and $\mu_C, \sigma_C = f_{encoder}(\mathcal{D}_C)$. Conditional variants (Garnelo et al., 2018a) instead compute a deterministic encoding, i.e., $z_C = f_{encoder}(\mathcal{D}_C)$.

**Querying:** In the querying phase, given target datapoints $x_T$ to make predictions for, the NP models the predictive distribution $p(y_T|x_T, z_C)$.

**Updating:** In the updating phase, the model receives new datapoints $\mathcal{D}_U$, and new encodings are computed, i.e., re-computing $z_C$ given $\mathcal{D}_C \leftarrow \mathcal{D}_C \cup \mathcal{D}_U$.

The deterministic variant maximizes the log-likelihood directly. In contrast, the stochastic variant maximizes an evidence lower bound (ELBO) of the log-likelihood instead:

$$\log p(y_T|x_T, \mathcal{D}_C) \geq \mathbb{E}_{q(z|\mathcal{D}_C \cup \mathcal{D}_T)} \left[\log p(y_T|x_T, z)\right] - \mathrm{KL}(q(z|\mathcal{D}_C \cup \mathcal{D}_T)||p(z|\mathcal{D}_C))$$

## 3 METHODOLOGY

In this section, we introduce the Constant Memory Attention Block (CMAB), a novel attention mechanism which preserves permutation invariance while only requiring (1) constant memory to compute its output and (2) constant computation to perform updates. Leveraging the efficiency properties of CMAB, we propose Constant Memory Attentive Neural Processes (CMANPs). We also introduce CMANP-AND (Autoregressive Not-Diagonal) extensions which only require constant memory in contrast to the quadratic memory required by prior Not-Diagonal extensions, allowing for scalability to a larger number of datapoints.

### 3.1 CONSTANT MEMORY ATTENTION BLOCK (CMAB)

Constant Memory Attention Block (Figure 1) takes as input the input data $\mathcal{D}$ and a set of input latent vectors $L_I$ and outputs a set of output latent vectors $L_I'$. The objective of the block is to encode the information of the input data $\mathcal{D}$ into a fixed-sized representation $|L_I|$. Each CMAB comprises two cross-attention modules, two self-attention modules, and one set of block-wise latent vectors $L_B$ whose value is learned during training. When stacking CMABs, the output latent vectors of the previous CMAB are fed as the input latent vectors to the next, i.e., $L_I \leftarrow L_I'$. Similar in style to that of iterative attention (Jaegle et al., 2021), the value of $L_I$ of the first stacked CMAB block is learned. A fundamental difference, however, is that iterative attention can neither compute the output in constant memory nor perform the updates in constant computation.

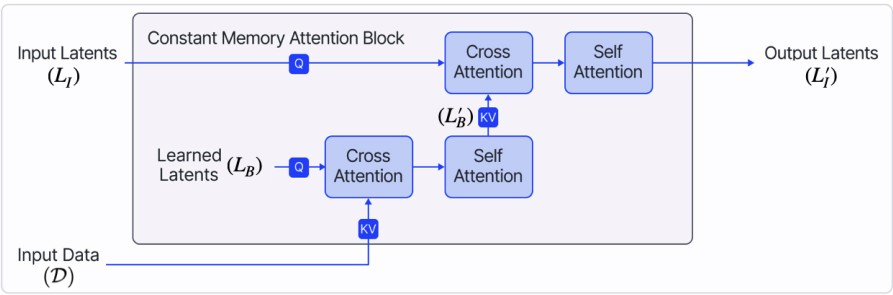

Figure 1: Constant Memory Attention Block (CMAB).

CMAB initially compresses the input data by applying a cross-attention between the input data and the block-wise latent vectors $L_B$. Next, self-attention is used to compute higher-order information:

$$L'_B = \text{SelfAttention}(\text{CrossAttention}(L_B, \mathcal{D}))$$

Afterwards, another cross-attention between the input vectors $L_I$ and $L'_B$ is performed and an additional self-attention is used to further compute higher-order information, resulting in the output vectors $L'_I$:

$$L'_I = \text{SelfAttention}(\text{CrossAttention}(L_I, L'_B))$$

In summary, CMAB (Figure 1) works as follows:

$$\textbf{CMAB}(L_I, \mathcal{D}) = L'_I = \textbf{SA}(\ \textbf{CA}(\ L_I,\ \text{SA}(\ \text{CA}(\ L_B, \mathcal{D}\ ))\ ))$$

where SA represents SelfAttention and CA represents CrossAttention. The two cross-attentions have a linear computational complexity of $\mathcal{O}(|\mathcal{D}||L_B|)$ and a constant computational complexity $\mathcal{O}(|L_B||L_I|)$. The self-attentions have constant complexities of $\mathcal{O}(|L_B|^2)$ and $\mathcal{O}(|L_I|^2)$. As such, the total computation required to produce the output of the block is $\mathcal{O}(|\mathcal{D}||L_B|+|L_B|^2+|L_B||L_I|+|L_I|^2)$ where the number of latents $|L_B|$ and $|L_I|$ are hyperparameter constants which bottleneck the amount of information which can be encoded.

### 3.1.1 Constant Computation Updates

A significant advantage of CMABs is that when given new inputs, CMABs can compute the updated output[1] in constant computation per new datapoint. In contrast, a transformer block would require re-computing its output from scratch, requiring quadratic computation to perform a similar update.

> Having previously computed $\textbf{CMAB}(L_I, \mathcal{D})$ and given new datapoints $\mathcal{D}_U$ (e.g., from sequential settings such as contextual bandits), $\textbf{CMAB}(L_I, \mathcal{D} \cup \mathcal{D}_U)$ can be computed in $\mathcal{O}(|\mathcal{D}_U|)$, i.e., a constant amount of computation per new datapoint.

**Proof Outline:** Since $|L_B|$ and $|L_I|$ are constants (hyperparameters), CMAB's complexity is constant except for the contributing complexity part of the first attention block: CrossAttention$(L_B, \mathcal{D})$, which has a complexity of $\mathcal{O}(|\mathcal{D}||L_B|)$. As such, to achieve constant computation updates, it suffices that the updated output of this cross-attention can be updated in constant computation per datapoint. Simplified, CrossAttention$(L_B, \mathcal{D})$ is computed as follows:

$$\text{CrossAttention}(L_B, \mathcal{D}) = \text{softmax}(QK^T)V$$

where $K$ and $V$ are key-value matrices respectively that represent the embeddings of the input data $\mathcal{D}$, and $Q$ is the query matrix representing the embeddings of the block-wise latents $L_B$. When an update with $|\mathcal{D}_U|$ new datapoints occurs, $|\mathcal{D}_U|$ rows are added to the key, value matrices. Crucially, the query matrix is constant due to the block-wise latent vectors $L_B$ being a fixed set of latent vectors

---

[1]CMABs also allow for efficient removal of datapoints (and consequently edits as well) to the input data, but this is outside the scope of this work.

whose values are learned. As a result, the output of the cross-attention can be computed via a rolling average in $\mathcal{O}(|\mathcal{D}_U|)$. A formal proof and description of this process is included in the Appendix.

As a result, we have the following update function:

$$\text{CrossAttention}(L_B, \mathcal{D} \cup \mathcal{D}_U) = \text{UPDATE}(\mathcal{D}_U, \text{CrossAttention}(L_B, \mathcal{D}))$$

where the UPDATE operation has a complexity of $\mathcal{O}(|\mathcal{D}_U|)$. Each of the remaining self-attention and cross-attention blocks only requires constant computation. As such, CMAB can compute its updated output in $\mathcal{O}(|\mathcal{D}_U|)$, i.e., a constant amount of computation per new datapoint.

### 3.1.2 Computing Output in Constant Memory

Leveraging the efficient updates property, CMABs can compute their output in constant memory regardless of the number of inputs. Naive computation of the output of CMAB requires non-constant memory due to $\text{CrossAttention}(L_B, \mathcal{D})$ having a linear memory complexity of $\mathcal{O}(|\mathcal{D}||L_B|)$. To achieve constant memory computation, we split the input data $\mathcal{D}$ into $|\mathcal{D}|/b_C$ batches of input datapoints of size up to $b_C$ (a pre-specified constant), i.e., $\mathcal{D} = \cup_{i=1}^{|\mathcal{D}|/b_C} \mathcal{D}_i$. Instead of computing the output at once, it is equivalent to performing an update $|\mathcal{D}|/b_C - 1$ times:

$$\mathbf{CA}(L_B, \mathcal{D}) = \text{UPDATE}(\mathcal{D}_1, \text{UPDATE}(\mathcal{D}_2, \ldots \text{UPDATE}(\mathcal{D}_{|\mathcal{D}|/b_C - 1}, \mathbf{CA}(L_B, \mathcal{D}_{|\mathcal{D}|/b_C}))))$$

Computing $\text{CrossAttention}(L_B, \mathcal{D}_{|\mathcal{D}|/B_C})$ requires $\mathcal{O}(b_C|L_B|)$, i.e., constant memory since $b_C$ and $|L_B|$ are both constants. After its computation, the memory can be freed up, so that each of the subsequent UPDATE operations can re-use the memory space one by one. Each of the update operations also costs $\mathcal{O}(b_C|L_B|)$ constant memory, resulting in $\mathbf{CrossAttention}(L_B, \mathcal{D})$ only needing constant memory $\mathcal{O}(b_C|L_B|)$ in total. As a result, CMAB's output can be computed in constant memory.

### 3.1.3 Additional Useful Properties

Since CMABs leverage only cross-attention and self-attention modules where both are permutation invariant, **CMABs are also permutation invariant** by nature. Similar to transformers, CMABs can leverage positional encodings for sequence and temporal data. Another advantage of CMABs over prior attention works is that the original input data $\mathcal{D}$ does not need to be stored when performing updates, meaning the model has privacy-preserving properties and is applicable to streaming data settings where data cannot be stored. CMABs only require (1) constant memory regardless of the number of inputs, making them particularly useful for scaling to large amounts of inputs, and (2) constant computation to perform updates, making them particularly useful for settings where the data comes in a stream and updates need to be performed to the dataset (e.g., contextual bandits, bayesian optimization, active learning, and temporal data). The efficiency of CMABs allows for modern attention models to be highly accessible for low-compute domains (e.g, IoT devices). To showcase CMABs general applicability, we included in the Appendix a model for next-event prediction (Temporal Point Processes) that also leverages CMABs.

### 3.2 Constant Memory Attentive Neural Process (CMANP)

In this section, we introduce Constant Memory Attentive Neural Processes (CMANPs), a memory efficient variant of Neural Processes (Figure 2) based on CMAB blocks. The conditioning, querying, and updating phases in CMANPs work as follows:

**Conditioning Phase:** In the conditioning phase, the CMAB blocks encode the context dataset into a set of latent vectors $L_i$. The first block takes as input a set of meta-learned latent vectors $L_0$ (i.e., $L_I$ in CMABs) and the context dataset $\mathcal{D}_C$, and outputs a set of encodings $L_1$ (i.e., $L'_I$ in CMABs). The output latents of each block are passed as the input latents to the next CMAB block.

$$L_i = \mathbf{CMAB}(L_{i-1}, \mathcal{D}_C)$$

Since CMAB can compute its output in constant memory, CMANPs can also perform this conditioning phase in constant memory.

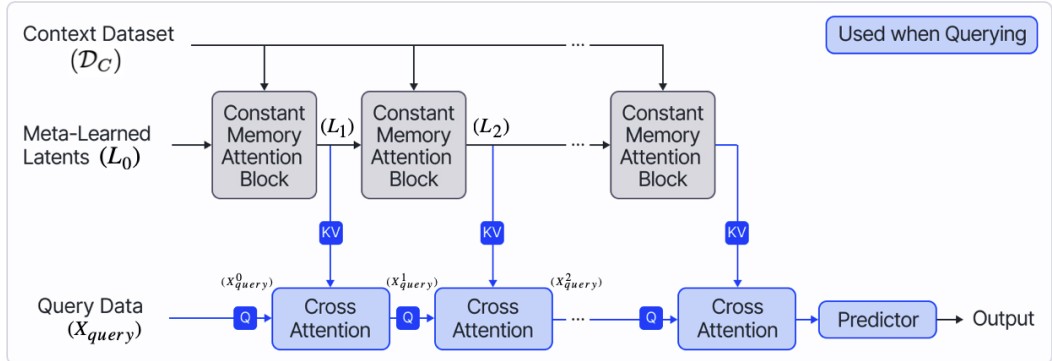

Figure 2: Constant Memory Attentive Neural Processes.

|  | Memory Complexity | | | | |
|---|---|---|---|---|---|
|  | **Conditioning** | **Querying** | | **Updating** | |
| **In Terms of** | $|\mathcal{D}_C|$ | $|\mathcal{D}_C|$ | $M$ | $|\mathcal{D}_C|$ | $|\mathcal{D}_U|$ |
| **TNP-D** | N/A | ✗ | ✗ | N/A | N/A |
| **TNP-ND** | N/A | ✗ | ✗ | N/A | N/A |
| **EQTNP** | ✗ | ◐ | ◐ | ✗ | ✗ |
| **LBANP** | ◐ | ✓ | ◐ | ◐ | ◐ |
| **LBANP-ND** | ◐ | ✓ | ✗ | ◐ | ◐ |
| **CMANP (Ours)** | ✓ | ✓ | ◐ | ✓ | ✓ |
| **CMANP-AND (Ours)** | ✓ | ✓ | ◐ | ✓ | ✓ |

Table 1: Comparison of Memory Complexities of top-performing Neural Processes with respect to the number of context datapoints $|\mathcal{D}_C|$, number of target datapoints in a batch $M$, and the number of new datapoints in an update $|\mathcal{D}_U|$. (Green) Checkmarks represent requiring constant memory, (Orange) half checkmarks represent requiring linear memory, and (Red) Xs represent requiring quadratic or more memory. A larger table with all baselines is included in the Appendix.

**Querying Phase:** In the querying phase, the deployed model retrieves information from the fixed size outputs of the CMAB blocks ($L_i$) to make predictions for the query datapoints ($X_{query}$). Beginning with $X^0_{query} \leftarrow X_{query}$, when making a prediction for the query datapoints $X_{query}$, information is retrieved via cross-attention.

$$X^i_{query} = \text{CrossAttention}(X^{i-1}_{query}, L_i)$$

**Update Phase:** In the update phase, the NP receives a batch of new datapoints $\mathcal{D}_U$ to include in the context dataset. CMANPs leverage the efficient update mechanism of CMABs to achieve efficient updates (constant per datapoint) to its context dataset, i.e., computing updated latents $L^{updated}_i$ given the new datapoints $\mathcal{D}_U$. Beginning with $L^{updated}_0 \leftarrow L_0$, the CMAB blocks are updated sequentially using the updated output of the previous CMAB block as follows:

$$L^{updated}_i = \textbf{CMAB}(L^{updated}_{i-1}, \mathcal{D}_C \cup \mathcal{D}_U)$$

Since CMAB can compute the output and perform updates in constant memory irrespective of the number of context datapoints, CMANPs can also compute its output and perform updates in constant memory. In Table 1, we compare the memory complexities of top-performing Neural Processes, showcasing the efficiency gains of CMANP over prior state-of-the-art methods.

### 3.2.1 AUTOREGRESSIVE NOT-DIAGONAL EXTENSION

In many settings where NPs are applied such as Image Completion, the target datapoints are correlated and their predictive distribution are evaluated altogether. As such, prior works (Nguyen & Grover, 2022; Feng et al., 2023) have proposed a Not-Diagonal variant of NPs which predicts the

mean and a full covariance matrix, typically via a low-rank approximation. This is in contrast to the vanilla (Diagonal) variants which predict the mean and a diagonal covariance matrix. Not-Diagonal methods, however, are not scalable, requiring quadratic memory in the number of target datapoints due to outputting a full covariance matrix.

Leveraging the efficient updates property of CMABs, we propose CMANP-AND (Autoregressive Not-Diagonal). During training, CMANP-AND follows the framework of prior Not-Diagonal variants. When deployed, the model is treated as an autoregressive model that makes predictions in blocks of size $b_Q$ datapoints. For each block prediction, a mean and full covariance matrix is computed via a low-rank approximation. Sampled predictions of prior blocks are used to make predictions for later blocks. The first block is sampled as follows: $\hat{y}_{N+1:N+b_Q} \sim \mathcal{N}(\mu_\theta(\mathcal{D}_C^0, x_{N+1:N+b_Q}), \Sigma_\theta(\mathcal{D}_C^0, x_{N+1:N+b_Q}))$. Afterwards, by leveraging the efficient update mechanism, CMANP-AND performs an update using the predictions $\{(x_i, \hat{y}_i)\}_{N+1}^{N+b_Q}$ as new context datapoints, meaning that CMANP-AND is now conditioned on a new context dataset $\mathcal{D}_C^1$ where $\mathcal{D}_C^1 = \mathcal{D}_C^0 \cup \{(x_i, \hat{y}_i)\}_{N+1}^{N+b_Q}$. Formally, the general formulation is as follows:

$$\hat{y}_{N+kb_Q+1:N+(k+1)b_Q} \sim \mathcal{N}(\mu_\theta(\mathcal{D}_C^k, x_{N+kb_Q+1:N+(k+1)b_Q}), \Sigma_\theta(\mathcal{D}_C^k, x_{N+kb_Q+1:N+(k+1)b_Q}))$$

where $k$ is the number of blocks already processed and $\mathcal{D}_C^k = \{(x_i, y_i)\}_{i=1}^N \cup \{(x_i, \hat{y}_i)\}_{N+1}^{N+kb_Q}$ is the context dataset. The hyperparameter $b_Q$ controls (1) the computational cost of the model in terms of memory and sequential computation length and (2) the performance of the model. Lower values of $b_Q$ allow for modelling more complex distributions, offering better performance but requiring more forward passes of the model. Since $b_Q$ is a constant, this Autoregressive Not-Diagonal extension makes predictions in constant memory, unlike prior Not-Diagonal variants which were quadratic in memory. As such, CMANP-AND can scale to a larger number of datapoints than prior methods (LBANP-ND and TNP-ND). The big-$\mathcal{O}$ complexity analysis is included in the Appendix.

## 4 EXPERIMENTS

In this section, we aim to evaluate the empirical performance of CMANPs and provide an analysis of CMANPs, showcasing their versatility. To do so, we compare CMANPs against a large variety of members of the Neural Process family on standard NP benchmarks: image completion and meta-regression. Specifically, we compare against Conditional Neural Processes (CNPs) (Garnelo et al., 2018a), Neural Processes (NPs) (Garnelo et al., 2018b), Bootstrapping Neural Processes (BNPs) (Lee et al., 2020), (Conditional) Attentive Neural Processes (C)ANPs (Kim et al., 2019), Bootstrapping Attentive Neural Processes (BANPs) (Lee et al., 2020), Latent Bottlenecked Attentive Neural Processes (LBANPs) (Feng et al., 2023), and Transformer Neural Processes (TNPs) (Nguyen & Grover, 2022). We also compare against the Not-Diagonal variants of the state-of-the-art methods (LBANP-ND and TNP-ND). Notably, our proposed CMANPs leverage CMABs, LBANPs (Feng et al., 2023) leverage iterative attention (Jaegle et al., 2021), and TNPs leverage transformers (Vaswani et al., 2017).

For the purpose of consistency, we set the number of latents (i.e., bottleneck size) $|L_I| = |L_B| = 128$ across all experiments. We also set $b_Q = 5$. To fairly compare iterative attention and CMABs, we report results for LBANPs with the same sized bottleneck (i.e., number of latents $L = 128$) as CMANPs across all experiments. We later show in the analysis section (Section 4.2.1) that our reported performance of CMANPs can be further improved by increasing the number of latents ($|L_I|$ or $|L_B|$) and decreasing the prediction block size $b_Q$.

Due to space limitations, several details are included in the appendix (1) experiments on contextual multi-armed bandits with a setting where data comes in a stream; (2) implementation details[2] such as hyperparameters and their selection; and (3) an application of CMABs on Temporal Point Processes, showing CMABs' general applicability.

### 4.1 IMAGE COMPLETION

In this setting, we consider the image completion setting with two datasets: EMNIST (Cohen et al., 2017) and CelebA (Liu et al., 2015). The model is given a set of pixel values of an image and aims

---

[2]The code will be released upon acceptance.

| Method | CelebA | | | EMNIST | |
|---|---|---|---|---|---|
| | 32x32 | 64x64 | 128x128 | Seen (0-9) | Unseen (10-46) |
| CNP | 2.15 ± 0.01 | 2.43 ± 0.00 | 2.55 ± 0.02 | 0.73 ± 0.00 | 0.49 ± 0.01 |
| CANP | 2.66 ± 0.01 | 3.15 ± 0.00 | — | 0.94 ± 0.01 | 0.82 ± 0.01 |
| NP | 2.48 ± 0.02 | 2.60 ± 0.01 | 2.67 ± 0.01 | 0.79 ± 0.01 | 0.59 ± 0.01 |
| ANP | 2.90 ± 0.00 | — | — | 0.98 ± 0.00 | 0.89 ± 0.00 |
| BNP | 2.76 ± 0.01 | 2.97 ± 0.00 | — | 0.88 ± 0.01 | 0.73 ± 0.01 |
| BANP | 3.09 ± 0.00 | — | — | 1.01 ± 0.00 | 0.94 ± 0.00 |
| TNP-D | 3.89 ± 0.01 | 5.41 ± 0.01 | — | **1.46 ± 0.01** | **1.31 ± 0.00** |
| LBANP | 3.97 ± 0.02 | 5.09 ± 0.02 | 5.84 ± 0.01 | 1.39 ± 0.01 | 1.17 ± 0.01 |
| CMANP (Ours) | 3.93 ± 0.05 | 5.02 ± 0.14 | 5.55 ± 0.01 | 1.36 ± 0.01 | 1.09 ± 0.01 |
| TNP-ND | 5.48 ± 0.02 | — | — | **1.50 ± 0.00** | **1.31 ± 0.00** |
| LBANP-ND | 5.57 ± 0.03 | — | — | 1.42 ± 0.01 | 1.14 ± 0.01 |
| CMANP-AND (Ours) | **6.31 ± 0.04** | **6.96 ± 0.07** | **7.15 ± 0.14** | **1.48 ± 0.03** | 1.19 ± 0.03 |

Table 2: Image Completion Experiments. Each method is evaluated with 5 different seeds according to the log-likelihood (higher is better). The "dash" represents methods that could not be run because of the large memory requirement.

to predict the remaining pixels of the image. Each image corresponds to a unique function (Garnelo et al., 2018b). In this experiment, the $x$ values are rescaled to [-1, 1] and the $y$ values are rescaled to $[-0.5, 0.5]$. For each task, a randomly selected set of pixels are selected as context datapoints and target datapoints.

EMNIST comprises black and white images of handwritten letters of $32 \times 32$ resolution. 10 classes are used for training. The context and target datapoints are sampled according to $N \sim \mathcal{U}[3, 197]$ and $M \sim \mathcal{U}[3, 200 - N]$ respectively. CelebA comprises coloured images of celebrity faces. Methods are evaluated on various resolutions to show the scalability of the methods. In CelebA32, images are downsampled to $32 \times 32$ and the number of context and target datapoints are sampled according to $N \sim \mathcal{U}[3, 197]$ and $M \sim \mathcal{U}[3, 200-N]$ respectively. In CelebA64, the images are down-sampled to $64 \times 64$ and $N \sim \mathcal{U}[3, 797]$ and $M \sim \mathcal{U}[3, 800 - N]$. In CelebA128, the images are down-sampled to $128 \times 128$ and $N \sim \mathcal{U}[3, 1597]$ and $M \sim \mathcal{U}[3, 1600 - N]$.

**Results.** Although all NP baselines (see Table 2) were able to be evaluated on CelebA (32 x 32) and EMNIST, many were not able to scale to CelebA (64 x 64) and CelebA (128 x 128). All Not-Diagonal variants were not able to be trained on CelebA (64 x 64) and CelebA (128 x 128) due to being too computationally expensive and requiring quadratic computation and memory. In contrast, CMANP(-AND) was not affected by this limitation, showing empirically CMANP-AND is scalable to more datapoints than prior Not-Diagonal variants. The results show that CMANP-AND achieves clear state-of-the-art results on CelebA (32x32), CelebA (64x64), and CelebA (128x128). Furthermore, CMANP-AND achieves results competitive with state-of-the-art on EMNIST.

Notably, the vanilla variants of CMANP (CMAB-based model) and LBANP (iterative attention-based model (Jaegle et al., 2021)) achieve similar performance while having the same sized bottleneck, i.e., the number of latents in both baselines is 128. These results suggest that the improved efficiency properties (constant memory and constant computation updates) of CMABs come at little cost in performance compared to iterative attention.

## 4.2 1-D REGRESSION

In this experiment, the goal is to model an unknown function $f$ and make predictions for a batch of $M$ target datapoints given a batch of $N$ context datapoints. During each training epoch, a batch of $B = 16$ functions are sampled from a GP prior with an RBF kernel $f_i \sim GP(m, k)$ where $m(x) = 0$ and $k(x, x') = \sigma_f^2 \exp(\frac{-(x-x')^2}{2l^2})$. The hyperparameters are sampled according to $l \sim \mathcal{U}[0.6, 1.0]$, $\sigma_f \sim \mathcal{U}[0.1, 1.0]$, $N \sim \mathcal{U}[3, 47]$, and $M \sim \mathcal{U}[3, 50 - N]$. After training, the models are evaluated according to the log-likelihood of the targets on functions sampled from GPs with RBF and Matern 5/2 kernels.

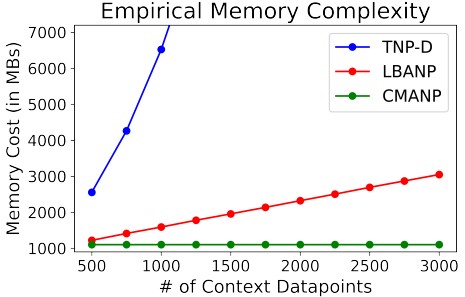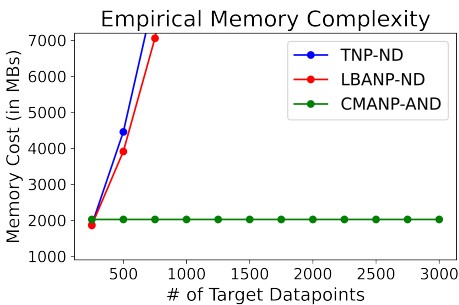

Figure 3: (Left) Comparison of memory usage of state-of-the-art NPs relative to the number of context datapoints. (Right) Comparison of memory usage of state-of-the-art NPs relative to the number of target datapoints.

**Results.** As shown in Table 3, CMANP-AND outperforms all baselines (except for TNP-ND) by a significant margin. CMANP-AND achieves comparable results to TNP-ND while only requiring constant memory. Once again, we see that the vanilla version of CMANP (CMAB-based model) and LBANP (iterative attention-based model (Jaegle et al., 2021)) achieve similar performance, further suggesting that CMABs' improves upon iterative attention in terms of efficiency (constant memory and constant computation updates) at little cost in performance.

### 4.2.1 ANALYSIS

**Empirical Memory:** Figure 3 compares the empirical memory cost of various state-of-the-art NP methods during evaluation. Comparing the vanilla variants of NPs, we see that TNP-D (transformer-based model) scales quadratically with respect to the number of context datapoints while LBANP (iterative attention-based model) scales linearly. In contrast, CMANP (CMAB-based model) only requires a low constant amount of memory regardless of the number of context datapoints. Comparing the Not-Diagonal variant of NPs, we see that both TNP-ND and LBANP-ND scale quadratically with respect to the number of target datapoints, limiting their applications. In contrast, CMANP-AND can scale to a far

| Method | RBF | Matern 5/2 |
|---|---|---|
| CNP | $0.26 \pm 0.02$ | $0.04 \pm 0.02$ |
| CANP | $0.79 \pm 0.00$ | $0.62 \pm 0.00$ |
| NP | $0.27 \pm 0.01$ | $0.07 \pm 0.01$ |
| ANP | $0.81 \pm 0.00$ | $0.63 \pm 0.00$ |
| BNP | $0.38 \pm 0.02$ | $0.18 \pm 0.02$ |
| BANP | $0.82 \pm 0.01$ | $0.66 \pm 0.00$ |
| TNP-D | $1.39 \pm 0.00$ | $0.95 \pm 0.01$ |
| LBANP | $1.27 \pm 0.02$ | $0.85 \pm 0.02$ |
| CMANP (Ours) | $1.24 \pm 0.01$ | $0.80 \pm 0.01$ |
| TNP-ND | $\mathbf{1.46 \pm 0.00}$ | $\mathbf{1.02 \pm 0.00}$ |
| LBANP-ND | $1.24 \pm 0.03$ | $0.78 \pm 0.02$ |
| CMANP-AND (Ours) | $\mathbf{1.48 \pm 0.03}$ | $0.96 \pm 0.01$ |

Table 3: 1-D Meta-Regression Experiments with log-likelihood metric (higher is better).

larger number of target datapoints. As a result, we can note that CMANPs are significantly more memory efficient and scalable to more datapoints than prior state-of-the-art methods.

**Effect of $b_Q$:** Figure 4 compares performance with respect to varying query block sizes $b_Q$ for CMANP-AND. We see that smaller block sizes achieve significantly better performance. This is expected as the autoregressive nature of the Neural Process results in a more flexible predictive distribution and hence better performance at the cost of an increased time complexity. We provide an analysis of the time complexity in the appendix (Figures 6 and 7).

**Varying Number of Latents** In Figure 4, we evaluated the result of varying the number of input latents ($L_I$) and the number of latents per block ($L_B$). We found that increasing the size of the bottleneck (i.e., number of latents $L_I$ and $L_B$) considerably improves the performance of the model. This, however, naturally comes at an increased memory cost.

## 5 RELATED WORK

Transformers (Vaswani et al., 2017) have achieved a large amount of success in a wide range of applications. However, the quadratic scaling of Transformers limits their applications. As such, there

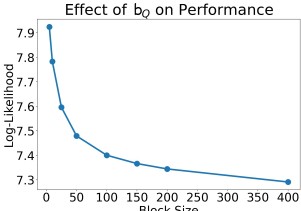 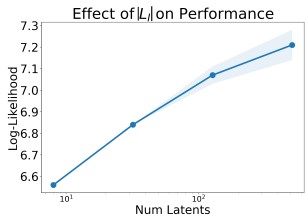 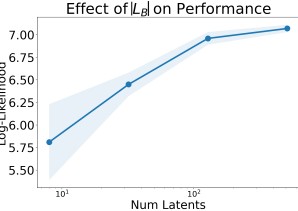

Figure 4: (Left) CMANP's performance relative to the size of the predictive block size ($b_Q$). (Middle) CMANP's performance relative to the number of input latent vectors ($|L_I|$). (Right) CMANP's performance relative to the number of block-wise latent vectors ($|L_B|$).

have been many follow-up works on efficient variants. However, very few works have achieved constant memory complexity. To the best of our knowledge, we are aware of only two works which have achieved a constant memory complexity. Rabe & Staats (2022) showed that self-attention can be computed in constant memory at the expense of an overall quadratic computation. Wu et al. (2022) proposed Memformer, a constant memory version of transformer specifically for sequence modelling problems by leveraging an external dynamic memory to encode and decode information that is updated over timesteps. As such, the memory/latent state of Memformer changes depending on the order of the datapoints. In contrast, CMABs only require linear computation, constant memory, and are by default permutation-invariant, i.e., not limited to sequence modelling. For an in-depth overview of follow-up works to Transformers, we refer the reader to the recent survey works (Khan et al., 2022; Lin et al., 2022).

Although CMABs have an efficient update mechanism reminiscent of RNNs (Cho et al., 2014; Chung et al., 2014; Hochreiter & Schmidhuber, 1997), their applications are different. RNNs are sensitive to input order, making their ideal setting applications which use sequential data. In contrast, by design, CMABs are by default permutation-invariant. Due to their long computation graph, RNNs also have issues such as vanishing gradients, making training these models with a large number of datapoints difficult. CMABs do not have issues with vanishing gradients since their ability to update efficiently is a fixed property of the module rather than RNN's learned mechanism.

NPs are applied in a wide range of applications which include Temporal Point Processes (Bae et al., 2023), sequence data (Singh et al., 2019; Willi et al., 2019), modelling stochastic physics fields (Holderrieth et al., 2021), robotics (Chen et al., 2022; Li et al., 2022b), and climate modeling (Vaughan et al., 2021). In doing so, there have been several methods proposed for encoding the context dataset. For example, CNPs (Garnelo et al., 2018a) encode the context set via a deep sets encoder (Zaheer et al., 2017), NPs (Garnelo et al., 2018b) propose to encode functional stochasticity via a latent variable. ConvCNPs (Gordon et al., 2019) use convolutions to build in translational equivariance. ANPs (Kim et al., 2019), LBANPs (Feng et al., 2023), and TNPs (Nguyen & Grover, 2022) use various kinds of attention. Recent work (Bruinsma et al., 2023) builds on CNPs and ConvCNPs by proposing to make them autoregressive at deployment. For an in-depth overview of NPs and their applications, we refer the reader to the recent survey work (Jha et al., 2022).

## 6 CONCLUSION

In this work, we introduced CMAB (Constant Memory Attention Block), a novel general-purpose attention block that (1) is permutation invariant, (2) computes its output in constant memory, and (3) performs updates in constant computation. Building on CMAB, we proposed Constant Memory Attentive Neural Processes (CMANPs), a new NP variant requiring only constant memory. Leveraging the efficient updates property of CMAB, we introduced CMANP-AND (Autoregressive Not-Diagonal extension). Empirically, we show that CMANP(-AND) achieves state-of-the-art results, while being significantly more efficient than prior state-of-the-art methods. In our analysis, we also showed that either by increasing the size of the latent bottleneck ($L_I$ and $L_B$) or decreasing the block size ($B_Q$), we can further improve the model performance.

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

## A APPENDIX: ADDITIONAL PROOF DETAILS

In this section, we (1) provide formal proof for CMAB's constant computation updates property, (2) include practical considerations to avoid numerical issues in the computation, (3) show that CMANPs uphold context and target invariance properties, and (4) include complexity analysis for CMANP-AND.

### A.1 CMAB'S CONSTANT COMPUTATION UPDATES PROOF

Recall, CMAB works as follows:

$$\mathbf{CMAB}(L_I, \mathcal{D}) = \mathbf{SA}(\mathbf{CA}(L_I, \mathbf{SA}(\mathrm{CA}(L_B, \mathcal{D}))))$$

where $\mathbf{SA}$ represents SelfAttention and $\mathbf{CA}$ represents CrossAttention. The two cross-attentions have a linear complexity of $\mathcal{O}(N|L_B|)$ and a constant complexity $\mathcal{O}(|L_B||L_I|)$, respectively where $N = |\mathcal{D}|$. The self-attentions have constant complexities of $\mathcal{O}(|L_B|^2)$ and $\mathcal{O}(|L_I|^2)$, respectively. As such, the total computation required to compute the output of the block is $\mathcal{O}(N|L_B| + |L_B|^2 + |L_B||L_I| + |L_I|^2)$ where $|L_B|$ and $|L_I|$ are hyperparameter constants which bottleneck the amount of information which can be encoded.

Importantly, since $|L_B|$ and $|L_I|$ are constants (hyperparameters), CMAB's complexity is constant except for the contributing complexity part of the first attention block: $\mathrm{CrossAttention}(L_B, \mathcal{D})$, which has a complexity of $\mathcal{O}(N|L_B|)$. To achieve constant computation updates, it suffices that the updated output of this cross-attention can be updated in constant computation per datapoint. Simplified, $\mathrm{CrossAttention}(L_B, \mathcal{D})$ is computed as follows:

$$\mathrm{emb} = \mathrm{CrossAttention}(L_B, \mathcal{D}) = \mathrm{softmax}(QK^T)V$$

where $K$ and $V$ are key, value matrices respectively that represent the embeddings of the context dataset $\mathcal{D}_C$ and $Q$ is the query matrix representing the embeddings of the block-wise latent vectors $L_B$. When an update with $\mathcal{D}_U$ new datapoints occurs, $|\mathcal{D}_U|$ rows are added to the key, value matrices. However, the query matrix is constant due to $L_B$ being a fixed set of latent vectors whose values are learned.

Without loss of generality, for simplicity, we consider the $j - th$ output vector of the cross-attention ($\mathrm{emb}_j$). Let $s_i = Q_{j,:}(K_{i,:})^T$ and $v_i = V_{i,:}$, then we have the following:

$$\mathrm{emb}_j = \sum_{i=1}^{N} \frac{\exp(s_i)}{C} v_i$$

where $C = \sum_{i=1}^{N} \exp(s_i)$. Performing an update with a set of new inputs $D_U$, results in adding $|\mathcal{D}_U|$ rows to the $K, V$ matrices:

$$\mathrm{emb}'_j = \sum_{i=1}^{N+|\mathcal{D}_U|} \frac{\exp(s_i)}{C'} v_i$$

where $C' = \sum_{i=1}^{N+|\mathcal{D}_U|} \exp(s_i) = C + \sum_{i=N+1}^{N+|\mathcal{D}_U|} \exp(s_i)$. As such, the updated embedding $\mathrm{emb}'_j$ can be computed via a rolling average:

$$\mathrm{emb}'_j = \frac{C}{C'} \times \mathrm{emb_j} + \sum_{i=N+1}^{N+|\mathcal{D}_U|} \frac{e^{s_i}}{C'} v_i$$

Computing $\mathrm{emb}'_j$ and $C'$ via this rolling average only requires $\mathcal{O}(|\mathcal{D}_U|)$ operations when given $C$ and $\mathrm{emb}$ as required. In practice, however, this is not stable. The computation can quickly run into numerical issues such as overflow problems.

**Practical Implementation:** In practice, instead of computing and storing $C$ and $C'$, we instead compute and store $\log(C)$ and $\log(C')$.

The update is instead computed as follows: $\log(C') = \log(C) + \mathrm{softplus}(T)$ where $T = \log(\sum_{i=N+1}^{N+|\mathcal{D}_U|} \exp(s_i - \log(C)))$. $T$ can be computed efficiently and accurately using the log-sum-exp trick in $\mathcal{O}(|\mathcal{D}_U|)$. This results in an update as follows:

$$\mathrm{emb}'_j = \exp(\log(C) - \log(C')) \times \mathrm{emb_j} + \sum_{i=N+1}^{N+|\mathcal{D}_U|} \exp(s_i - \log(C'))v_i$$

This method of implementation avoids the numerical issues that will occur while resulting in computing the same $emb'$. We detail how to derive the practical implementation below:

**Practical Implementation (Derivation):**

$$C = \sum_{i=1}^{N} \exp(s_i) \qquad C' = \sum_{i=1}^{N+|\mathcal{D}_U|} \exp(s_i)$$

$$\log(C') - \log(C) = \log(\sum_{i=1}^{N+|\mathcal{D}_U|} \exp(s_i)) - \log(\sum_{i=1}^{N} \exp(s_i))$$

$$\log(C') = \log(C) + \log(\frac{\sum_{i=1}^{N+|\mathcal{D}_U|} \exp(s_i)}{\sum_{i=1}^{N} \exp(s_i)})$$

$$\log(C') = \log(C) + \log(1 + \frac{\sum_{i=N+1}^{N+|\mathcal{D}_U|} \exp(s_i)}{\sum_{i=1}^{N} \exp(s_i)})$$

$$\log(C') = \log(C) + \log(1 + \frac{\sum_{i=N+1}^{N+|\mathcal{D}_U|} \exp(s_i)}{\exp(\log(C))})$$

$$\log(C') = \log(C) + \log(1 + \sum_{i=N+1}^{N+|\mathcal{D}_U|} \exp(s_i - \log(C)))$$

Let $T = \log(\sum_{i=N+1}^{N+|\mathcal{D}_U|} \exp(s_i - \log(C)))$. Note that $T$ can be computed efficiently using the log-sum-exp trick in $\mathcal{O}(|\mathcal{D}_U|)$. Also, recall the softplus function is defined as follows: $\mathrm{softplus}(k) = \log(1 + \exp(k))$. As such, we have the following:

$$\log(C') = \log(C) + \log(1 + \exp(T))$$
$$= \log(C) + \mathrm{softplus}(T)$$

Recall:

$$\mathrm{emb}'_j = \frac{C}{C'} \times \mathrm{emb_j} + \sum_{i=N+1}^{N+|\mathcal{D}_U|} \frac{\exp(s_i)}{C'} v_i$$

Re-formulating it using $\log(C)$ and $\log(C')$ instead of $C$ and $C'$ we have the following update:

$$\mathrm{emb}'_j = \exp(\log(C) - \log(C')) \times \mathrm{emb_j} + \sum_{i=N+1}^{N+|\mathcal{D}_U|} \exp(s_i - \log(C'))v_i$$

which only requires $\mathcal{O}(|\mathcal{D}_U|)$ computation (i.e., constant computation per datapoint) while avoiding numerical issues.

A.2    ADDITIONAL PROPERTIES

In this section, we show that CMANPs uphold the context and target invariance properties.

**Property: Context Invariance.** A Neural Process $p_\theta$ is context invariant if for any choice of permutation function $\pi$, context datapoints $\{(x_i, y_i)\}_{i=1}^N$, and target datapoints $x_{N+1:N+M}$,

$$p_\theta(y_{N+1:N+M}|x_{N+1:N+M}, x_{1:N}, y_{1:N}) = p_\theta(y_{N+1:N+M}|x_{N+1:N+M}, x_{\pi(1):\pi(N)}, y_{\pi(1):\pi(N)})$$

**Proof Outline:** Since CMANPs retrieve information from a compressed encoding of the context dataset computed by CMAB (Constant Memory Attention Block). It suffices to show that CMABs compute their output while being order invariant in their input (i.e., context dataset in CMANPs) ($\mathcal{D}$).

Recall CMAB's work as follows:

$$\mathbf{CMAB}(L_I, \mathcal{D}) = \mathbf{SA}(\mathbf{CA}(L_I, \mathbf{SA}(\mathbf{CA}(L_B, \mathcal{D}))))$$

where $L_I$ is a set of vectors outputted by prior blocks, $L_B$ is a set of vectors whose values are learned during training, and $\mathcal{D}$ are the set of inputs in which we wish to be order invariant in.

The first cross-attention to be computed is $\mathbf{CA}(L_B, \mathcal{D})$. A nice feature of cross-attention is that its order-invariant in the keys and values; in this case, these are embeddings of $\mathcal{D}$. In other words, the output of $\mathbf{CA}(L_B, \mathcal{D})$ is order invariant in the input data $\mathcal{D}$.

Since the remaining self-attention and cross-attention blocks take as input: $L_I$ and the output of $\mathbf{CA}(L_B, \mathcal{D})$, both of which are order invariant in $\mathcal{D}$, therefore the output of CMAB is order invariant in $\mathcal{D}$.

As such, CMANPs are also context invariant as required.

**Property: Target Equivariance.** A model $p_\theta$ is target equivariant if for any choice of permutation function $\pi$, context datapoints $\{(x_i, y_i)\}_{i=1}^N$, and target datapoints $x_{N+1:N+M}$,

$$p_\theta(y_{N+1:N+M}|x_{N+1:N+M}, x_{1:N}, y_{1:N}) = p_\theta(y_{\pi(N+1):\pi(N+M)}|x_{\pi(N+1):\pi(N+M)}, x_{1:N}, y_{1:N})$$

**Proof Outline:** The vanilla variant of CMANPs makes predictions similar to that of LBANPs (Feng et al., 2023) by retrieving information from a set of latent vectors via cross-attention and uses an MLP (Predictor). The architecture design of LBANPs ensure that the result is equivalent to making the predictions independently. As such, CMANPs preserve target equivariance the same way LBANPs do.

However, for the Autoregressive Not-Diagonal variant (CMANP-AND), the target equivariance is not held as it depends on the order in which the datapoints are processed. This is in common with that of prior methods by Nguyen & Grover (2022) and Bruinsma et al. (2023).

### A.3  COMPLEXITY ANALYSIS FOR CMANP-AND

For a batch of $M$ datapoints and a prediction block size of $b_Q$ (hyperparameter constant), there are $\lceil \frac{M}{b_Q} \rceil$ batches of datapoints whose predictions are made autoregressively. Each batch incurs a constant complexity of $\mathcal{O}(b_Q)^2$ due to predicting a full covariance matrix. As such for a batch of $M$ target datapoints, CMANP-AND requires a sub-quadratic total computation of $\mathcal{O}(\lceil \frac{M}{b_Q} \rceil b_Q^2) = \mathcal{O}(Mb_Q)$ with a sequential computation length of $\mathcal{O}(\frac{M}{b_Q})$. Crucially, CMANP-AND only requires constant memory in $|\mathcal{D}_C|$ and linear memory in $M$, making it significantly more efficient than prior works which required at least quadratic memory.

## B  APPENDIX: ADDITIONAL EXPERIMENTS AND ANALYSES

In this section, we (1) showcase the versatility of CMABs by applying them to Temporal Point Processes, (2) show results for CMANPs on Contextual Bandits, a setting where the amount of data increases over time, (3) include a memory complexity table which includes all baselines, and (4) analyse the time cost and performance relative to several hyperparameters.

|      | Mooc | | | Reddit | | |
|------|------|------|------|------|------|------|
|      | RMSE | NLL | ACC | RMSE | NLL | ACC |
| THP  | 0.202 ± 0.017 | **0.267 ± 0.164** | **0.336 ± 0.007** | **0.238 ± 0.028** | **0.268 ± 0.098** | **0.610 ± 0.002** |
| CMHP | **0.168 ± 0.011** | **-0.040 ± 0.620** | 0.237 ± 0.024 | 0.262 ± 0.037 | 0.528 ± 0.209 | 0.609 ± 0.003 |

Table 4: Temporal Point Processes Experiments.

## B.1 APPLYING CMABS TO TEMPORAL POINT PROCESSES (TPPS)

In this section, we highlight the effectiveness of our proposed Constant Memory Attention Block by applying it to settings beyond that of Neural Processes. Specifically, we apply CMABs to Temporal Point Processes (TPPs). In brief, Temporal Point Processes are stochastic processes composed of a time series of discrete events. Recent works have proposed to model this via a neural network. Notably, models such as THP (Zuo et al., 2020) encode the history of past events to predict the next event, i.e., modelling the predictive distribution of the next event $p_\theta(\tau_{l+1}|\tau_{\leq l})$ where $\theta$ are the parameters of the model, $\tau$ represents an event, and $l$ is the number of events that have passed. Typically, an event comprises a discrete temporal (time) stamp and a mark (categorical class).

### B.1.1 CONSTANT MEMORY HAWKES PROCESSES (CMHPS)

Building on CMABs, we introduce the Constant Memory Hawkes Process (CMHPs) (Figure 5), a model which replaced the transformer layers in Transformer Hawkes Process (THP) (Zuo et al., 2020) with Constant Memory Attention Blocks. However, unlike THPs which summarise the information for prediction in a single vector, CMHPs summarise it into a set of latent vectors. As such, a flattening operation is added at the end of the model. Following prior work (Bae et al., 2023; Shchur et al., 2020), we use a mixture of log-normal distribution as the decoder for both THP and CMHP.

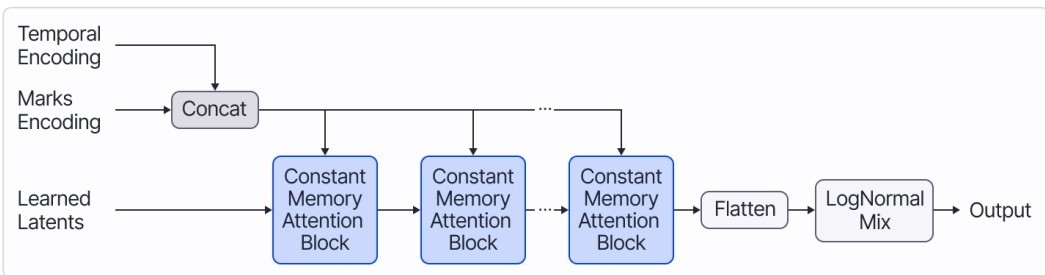

Figure 5: Constant Memory Hawkes Processes

### B.1.2 CMHPS: EXPERIMENTS

In this experiment, we compare CMHPs against THPs on standard TPP datasets: Mooc and Reddit.

**Mooc Dataset.** comprises of $7,047$ sequences. Each sequence contains the action times of an individual user of an online Mooc course with 98 categories for the marks.

**Reddit Dataset.** comprises of $10,000$ sequences. Each sequence contains the action times from the most active users with marks being one of the $984$ the subreddit categories of each sequence.

The results (Table 4) suggest that replacing the transformer layer with CMAB (Constant Memory Attention Block) results in a small drop in performance. Crucially, unlike THP, CMHP has the ability to efficiently update the model with new data as it arrives over time which is typical in time series data such as in Temporal Point Processes. CMHP only pays constant computation to update the model unlike the quadratic computation required by THP.

## B.2 ADDITIONAL CMANPS EXPERIMENTS: CONTEXTUAL BANDITS

In the Contextual Bandit setting introduced by Riquelme et al. (2018), a unit circle is divided into 5 sections which contain 1 low reward section and 4 high reward sections $\delta$ defines the size of the low

| Method | $\delta = 0.7$ | $\delta = 0.9$ | $\delta = 0.95$ | $\delta = 0.99$ | $\delta = 0.995$ |
|---|---|---|---|---|---|
| Uniform | $100.00 \pm 1.18$ | $100.00 \pm 3.03$ | $100.00 \pm 4.16$ | $100.00 \pm 7.52$ | $100.00 \pm 8.11$ |
| CNP | $4.08 \pm 0.29$ | $8.14 \pm 0.33$ | $8.01 \pm 0.40$ | $26.78 \pm 0.85$ | $38.25 \pm 1.01$ |
| CANP | $8.08 \pm 9.93$ | $11.69 \pm 11.96$ | $24.49 \pm 13.25$ | $47.33 \pm 20.49$ | $49.59 \pm 17.87$ |
| NP | $1.56 \pm 0.13$ | $2.96 \pm 0.28$ | $4.24 \pm 0.22$ | $18.00 \pm 0.42$ | $25.53 \pm 0.18$ |
| ANP | $1.62 \pm 0.16$ | $4.05 \pm 0.31$ | $5.39 \pm 0.50$ | $19.57 \pm 0.67$ | $27.65 \pm 0.95$ |
| BNP | $62.51 \pm 1.07$ | $57.49 \pm 2.13$ | $58.22 \pm 2.27$ | $58.91 \pm 3.77$ | $62.50 \pm 4.85$ |
| BANP | $4.23 \pm 16.58$ | $12.42 \pm 29.58$ | $31.10 \pm 36.10$ | $52.59 \pm 18.11$ | $49.55 \pm 14.52$ |
| TNP-D | $\mathbf{1.18 \pm 0.94}$ | $\mathbf{1.70 \pm 0.41}$ | $2.55 \pm 0.43$ | $\mathbf{3.57 \pm 1.22}$ | $\mathbf{4.68 \pm 1.09}$ |
| LBANP | $\mathbf{1.11 \pm 0.36}$ | $\mathbf{1.75 \pm 0.22}$ | $\mathbf{1.65 \pm 0.23}$ | $6.13 \pm 0.44$ | $8.76 \pm 0.15$ |
| CMANP (Ours) | $\mathbf{0.93 \pm 0.12}$ | $\mathbf{1.56 \pm 0.10}$ | $\mathbf{1.87 \pm 0.32}$ | $9.04 \pm 0.42$ | $13.02 \pm 0.03$ |

Table 5: Contextual Multi-Armed Bandit Experiments with varying $\delta$. Models are evaluated according to cumulative regret (lower is better). Each model is run 50 times for each value of $\delta$.

reward section while the 4 high reward sections have equal sizes. In each round, the agent has to select 1 of 5 arms that each represent one of the regions. For context during the selection, the agent is given a 2-D coordinate $X$ and the actions it selected and rewards it received in previous rounds.

If $||X|| < \delta$, then the agent is within the low reward section. If the agent pulls arm 1, then the agent receives a reward of $r \sim \mathcal{N}(1.2, 0.012)$. Otherwise, if the agent pulls a different arm, then it receives a reward $r \sim \mathcal{N}(1.0, 0.012)$. Consequently, if $||X|| \geq \delta$, then the agent is within one of the four high-reward sections. If the agent is within a high reward region and selects the corresponding arm to the region, then the agent receives a large reward of $N \sim \mathcal{N}(50.0, 0.012)$. Alternatively, pulling arm 1 will reward the agent with a small reward of $r \sim \mathcal{N}(1.2, 0.012)$. Pulling any of the other 3 arms rewards the agent with an even smaller reward of $r \sim \mathcal{N}(1.0, 0.012)$.

During each training iteration, $B = 8$ problems are sampled. Each problem is defined by $\{\delta_i\}_{i=1}^B$ which are sampled according to a uniform distribution $\delta \sim \mathcal{U}(0, 1)$. $N = 512$ points are sampled as context datapoints and $M = 50$ points are sampled for evaluation. Each datapoint comprises of a tuple $(X, r)$ where $X$ is the coordinate and $r$ is the reward values for the 5 arms. The objective of the model during training is to predict the reward values for the 5 arms given the coordinates (context datapoints).

During the evaluation, the model is run for 2000 steps. At each step, the agent selects the arm which maximizes its UCB (Upper-Confidence Bound). After which, the agent receives the reward value corresponding to the arm. The performance of the agent is measured by cumulative regret. For comparison, we evaluate the modes with varying $\delta$ values and report the mean and standard deviation for 50 seeds.

**Results.** In Table 5, we compare CMANPs with other NP baselines, including the recent state-of-the-art methods TNP-D, EQTNP, and LBANP. We see that CMANP achieves competitive performance with state-of-the-art for $\delta \in \{0.7, 0.9, 0.95\}$. However, the performance degrades as $\delta$ reaches extreme values close to the limit such as 0.99 and 0.995 – settings that are at the edge of the training distribution.

### B.3 ADDITIONAL ANALYSES

**Memory Complexity:** In Table 6, we include a comparison of CMANPs with all NP baselines, showing that CMANPs are amongst the best in terms of memory efficiency when compared to prior NP methods. Notably, the methods with a similar memory complexity to CMANPs perform significantly worse in terms of performance across the various experiments (Tables 2 and 3)). As such, CMANPs provide the best trade-off in terms of memory and performance.

**Time Cost and Performance Scatterplot:** In Figure 6, we evaluate the empirical time cost of CMANP-AND with varying number of context datapoints ($N = |\mathcal{D}_C|$), number of target datapoints ($M$), and block size ($b_Q$). The number of context datapoints and the number of target datapoints are shown as labels in the scatterplot. The colour of the points on the scatterplot represents its respective block size. Depending on the amount of available resources (e.g., time), the value of the block size can be chosen equivalently.

| In Terms of | Conditioning $|\mathcal{D}_C|$ | Querying $|\mathcal{D}_C|$ | Querying $M$ | Updating $|\mathcal{D}_C|$ | Updating $|\mathcal{D}_U|$ |
|---|---|---|---|---|---|
| CNP | ✓ | ✓ | ✓(half) | ✓ | ✓ |
| CANP | ✗ | ✗ | ✓(half) | ✗ | ✗ |
| NP | ✓ | ✓ | ✓(half) | ✓ | ✓ |
| ANP | ✗ | ✗ | ✓(half) | ✗ | ✗ |
| BNP | ✓ | ✓ | ✓(half) | ✓ | ✓ |
| BANP | ✗ | ✗ | ✓(half) | ✗ | ✗ |
| TNP-D | N/A | ✗ | ✗ | N/A | N/A |
| LBANP | ✓(half) | ✓ | ✓(half) | ✓(half) | ✓(half) |
| **CMANP (Ours)** | ✓ | ✓ | ✓(half) | ✓ | ✓ |
| TNP-ND | N/A | ✗ | ✗ | N/A | N/A |
| LBANP-ND | ✓(half) | ✓(half) | ✗ | ✓(half) | ✓(half) |
| **CMANP-AND (Ours)** | ✓ | ✓ | ✓(half) | ✓ | ✓ |

Table 6: Comparison of Memory Complexities of Neural Processes with respect to the number of context datapoints $|\mathcal{D}_C|$, number of target datapoints in a batch $M$, and the number of new datapoints in an update $|\mathcal{D}_U|$. (Green) Checkmarks represent requiring constant memory, (Orange) half checkmarks represent requiring linear memory, and (Red) Xs represent requiring quadratic or more memory. A table with all baselines are included in the Appendix.

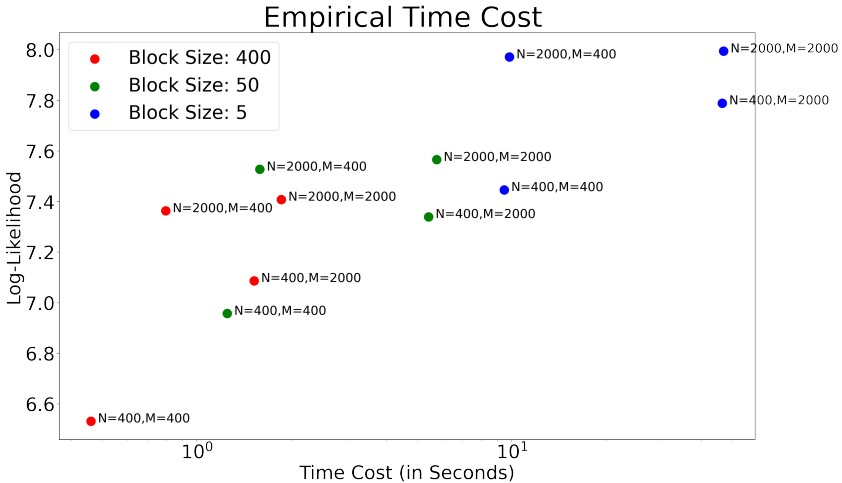

Figure 6: Scatterplot comparing the empirical time cost of CMANP-AND with respect to the block size ($b_Q$), number of context datapoints ($N$), and number of target datapoints ($M$).

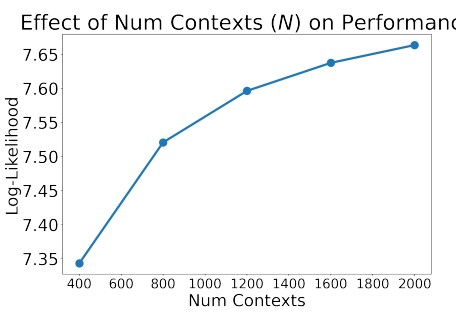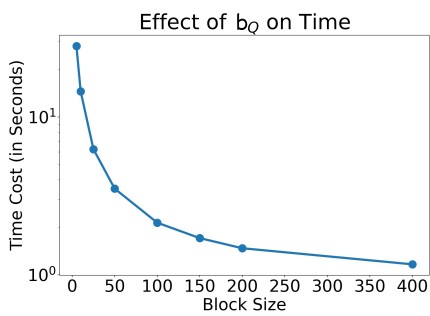

Figure 7: Additional Analyses Graphs

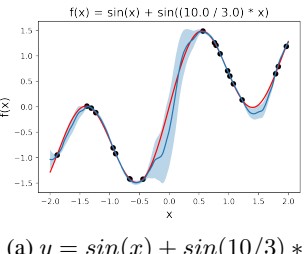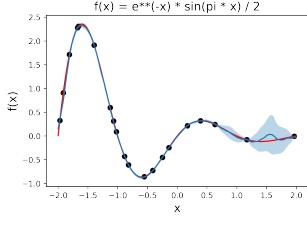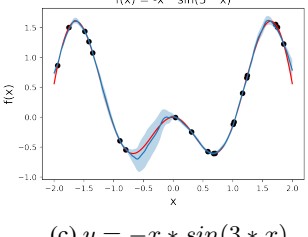

(a) $y = sin(x) + sin(10/3) * x$  (b) $y = e^{-x} * sin(\pi * x)/2$  (c) $y = -x * sin(3 * x)$

Figure 8: CMANPs 1-D Regression Visualizations

**Generalisation Ability:** In Figure 7, we evaluated CMANP-AND's potential to generalize to settings with significantly more context datapoints than originally trained on. During training, the model was trained on tasks with a maximum of 800 context datapoints. In contrast, during evaluation, we conditioned on up to 2000 context datapoints and evaluated on 800 target datapoints. Empirically, we found that the model's performance grows consistently as the number of context datapoints increases. However, the performance slows down at large number of contexts. We hypothesize that the cause of the saturation is due to two main factors: (1) the information gained from new context datapoints is dependent on the size of the current context dataset. For example, adding 400 new datapoints to a context dataset of size 400 results in 100% more data. Alternatively, adding 400 new datapoints to a context dataset of size 1600 results in 25% more data. As such, it is expected to see such saturation with a linear x-axis scaling. (2) in this case, CelebA (64 x 64) comprising of only 4096 pixels in total. 2000 comprises of a substantial amount of the data, i.e., approximately half. As such, saturation is expected as the amount of information gained by additional datapoints is minimal.

**Effect of Block Size ($b_Q$) on Empirical Time Cost:** In Figure 7, we evaluated the time required for CMANP-AND with respect to the block size ($b_Q$). The results are as expected, showing that the time required during deployment is lower as the block size increases. In the main paper, we showed that lower block sizes improve the model's performance. In conjunction, these plots show that there is a trade-off between the time cost and performance. These results suggest that during deployment it is advisable to select smaller block sizes if allowed for the time constraint.

**Visualizations:** In Figures 8 and 10, we show visualizations for the 1-D regression and Image Completion tasks respectively. Figure 9 show out-of-distribution visualizations where the context datapoints are only sampled from part of the distribution.

**Number of Latents Comparison with LBANPs:** A major factor that affects the performance in iterative attention-based models is the size of the bottleneck (i.e., the number of latents). Feng et al. (2023) showed that the performance of LBANPs (iterative attention based Neural Process) can change significantly depending on $|L_{LBANPs}|$ (the number of latents). As such, for the sake of fairness, in our paper, we similarly set the number of latents in CMANPs to match the same number of latents used in LBANPs' paper, i.e., $|L_I| = |L_B| = |L_{LBANPs}|$.

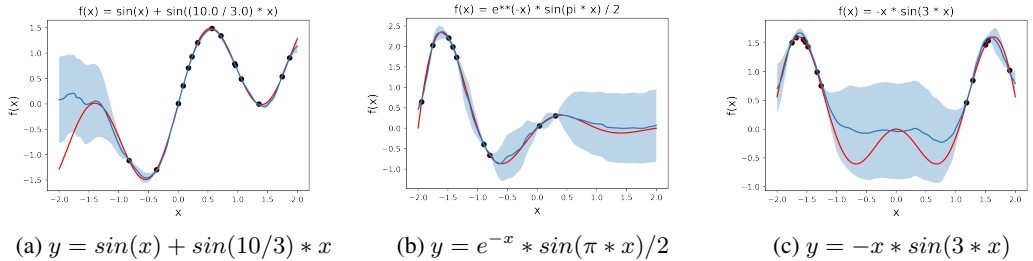

(a) $y = sin(x) + sin(10/3) * x$     (b) $y = e^{-x} * sin(\pi * x)/2$     (c) $y = -x * sin(3 * x)$

Figure 9: CMANPs 1-D Out-of-Distribution Regression Visualizations. The model is evaluated between $[-2.0, 2.0]$. However, context datapoints are sampled from only (a) $[-1.0, 2.0]$, (b) $[-2.0, 1.0]$, and (c) $[-2.0, -1.0] \cup [1.0, 2.0]$.

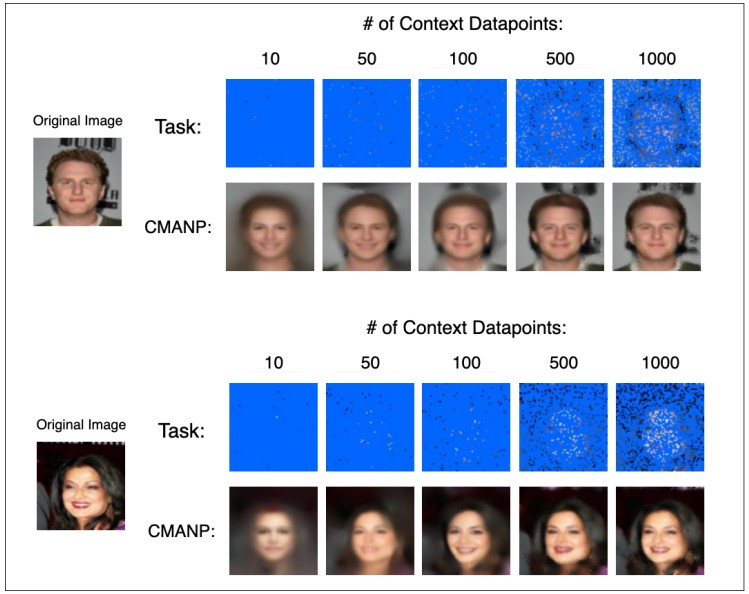

Figure 10: CMANPs Image Completion Visualizations

| Num Latents | CMANPs | LBANPs |
|---|---|---|
| 8 | $3.49 \pm 0.02$ | $3.54 \pm 0.01$ |
| 16 | $3.60 \pm 0.03$ | $3.64 \pm 0.02$ |
| 32 | $3.73 \pm 0.03$ | $3.77 \pm 0.01$ |
| 64 | $3.79 \pm 0.06$ | $3.88 \pm 0.01$ |
| 128 | $3.92 \pm 0.03$ | $3.97 \pm 0.02$ |

Table 7: Comparison of CMANPs with LBANPs for varying number of latents on the CelebA (32x32) image completion task. The number of latents in CMANPs matches the same number of latents used in LBANPs' paper, i.e., $|L_I| = |L_B| = |L_{LBANPs}|$. We see that CMANPs are competitive with LBANPs performing slightly worse. However, unlike LBANPs, CMANPs (1) computes their output in constant memory, and (2) perform updates in constant computation given new context tokens (in this case, pixels)

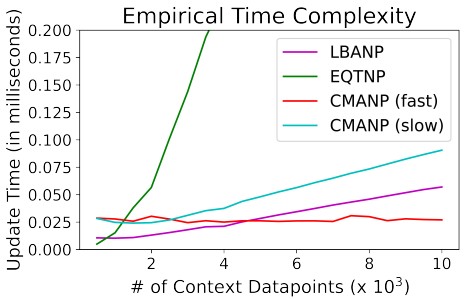
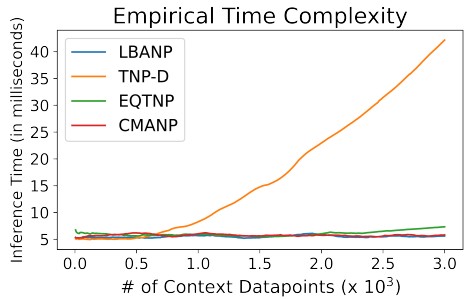

(a) Runtime analysis of the update process.

(b) Runtime analysis of the query process.

Figure 11: Analyses Graphs comparing the runtime of CMANPs with various baselines. (a) Comparison of the update procedure of CMAB-based NP (CMANPs) with Perceiver's iterative attention-based NP model (LBANPs) and a transformer-based NP model (EQTNP). CMANP (fast) refers to the CMAB's efficient update mechanism. CMANP (slow) refers to the traditional update mechanism. (b) Comparison of the query/inference process of CMANPs with LBANPs (Perceiver's iterative attention-based model), TNPs (Transformer-based model), and EQTNPs (Transformer-based model with an efficient query mechanism).

For completeness, we have included in Table 7 a comparison of the performance of CMANPs and LBANPs for varying number of latents for the CelebA (32x32) image completion task. We see that CMANPs are competitive with LBANPs performing slightly worse. However, unlike LBANPs, CMANPs (1) computes their output in constant memory, and (2) perform updates in constant computation given new context tokens (in this case, pixels).

**Empirical Time Comparison with Baselines:** In Figure 11a, we compare CMANP using the efficient update process with CMANP using the traditional update process, showing that the efficient update process is initially similar in runtime to the traditional update process. However, as the number of context datapoints increases (i.e., updates are performed) over time, the traditional update process requires linear runtime while our proposed efficient update process still only requires constant runtime.

In Figure 11a, we also compare the runtime of the update process of CMAB-based NP (CMANPs) with Perceiver's iterative attention-based NP model (LBANPs) and a transformer-based NP model. We see that the CMAB-based model only requires a constant amount of time to perform the update. In contrast, Perceiver's iterative attention-based model's update runtime scales linearly and Transformer model's update runtime scales quadratically.

In Figure 11b, we compare the querying (inference) runtime of CMANP with LBANPs (Perceiver's iterative attention-based model), TNPs (Transformer-based model). We see that CMANPs and LBANPs stay constant while the transformer-based model (TNP) scales quadratically in runtime.

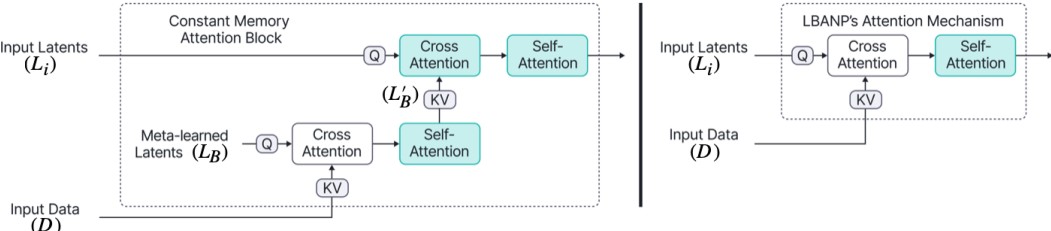

Figure 12: Comparison of our proposed Constant Memory Attention Block and that of LBANP's Attention Block (i.e., Perceiver's iterative attention). The green blocks indicate constant complexity. Naively computing the outputs, the white blocks indicate linear complexity. CMAB, however, can compute its white cross attention block in constant memory via a rolling average. LBANP's Attention block (Perceiver's iterative attention) cannot compute their white cross attention block in constant memory.

We would like to note, however, that runtime is highly dependent on the efficiency of the implementation and the hardware. Since our work focused primarily on the memory aspect rather than runtime, our implementation was that of a simple sequential version of CMABs and CMANPs. However, CMANPs have an architecture which allows for several modules within CMABs to be parallelized when performing updates for improved runtime. As such, we expect that an optimized codebase will be able to significantly improve CMAB's and CMANP's runtime.

## C  APPENDIX: DISCUSSION

In this section, we (1) compare Perceiver's iterative attention with CMABs, detailing why Perceiver cannot achieve the efficiency properties of CMAB, (2) compare the likelihood computation of Autoregressive Not-Diagonal extension with Not-Diagonal extension, and (3) compare NPs with other existing methods for uncertainty estimation.

### C.1  COMPARISON OF ITERATIVE ATTENTION WITH CMABS

Figure 12 compares Perceiver's iterative attention (used in LBANPs) with CMABs (used in CMANPs). In this subsection, we detail why Perceiver's iterative attention cannot achieve computing its output in constant memory and performing updates in constant computation. Notably, the property of constant memory is dependent on constant computation updates. Below, we detail why Perceiver's iterative attention does not have the constant computation updates property. Previously, we proved that the output of CrossAttention can be updated in constant computation per datapoint via a rolling summation given that the query vectors are constants. The efficiency gains revolve around CMABs' block-wise learnable latent vectors denoted as $L_B$ being a learned constant.

When stacked, CMABs work as follows: $L_{i+1} = \text{SA}(\text{CA}(L_i, L'_B))$ where $L'_B = \text{SA}(\text{CA}(L^i_B, \mathcal{D}))$ and $L^i_B$ denote the block-wise latent vectors for the $i$-th CMAB.

Perceiver's iterative attention block works as follows: $L_{i+1} = \text{SA}(\text{CA}(\text{L}_i, \mathcal{D}))$.

When new datapoints $\mathcal{D}_U$ is added to the input, i.e., $\mathcal{D} \leftarrow \mathcal{D} \cup \mathcal{D}_U$, the input latents ($L^{updated}_i \neq L_i$ where $i > 0$) change and is thus not a constant. As such, Perceiver's iterative attention do not allow for (1) constant computation updates and (2) computing output in constant memory, making it more expensive in terms of memory compared to CMABs.

For CMABs, computing $L_{i+1} = \text{SA}(\text{CA}(L_i, L'_B))$ is always constant in computation since $|L_i|$ and $|L'_B|$ are constant in size. Computing the updated output: $L^i_B = \text{SA}(\text{CA}(L_B, \mathcal{D} \cup D_U))$ can always be computed in constant computation because $L_B$ is a constant.

## C.2 COMPARISON OF THE LIKELIHOOD COMPUTATION OF AUTOREGRESSIVE NOT-DIAGONAL EXTENSION WITH NOT-DIAGONAL EXTENSION:

In brief, the Autoregressive Not-Diagonal extension is different from Not-Diagonal extension in that the predictions are made autoregressively which allows for more flexible distributions than prior Not-Diagonal variants. As such, it is expected that the autoregressive not-diagonal variant's likelihood is higher than that of the non-autoregressive baselines which only model an unimodal gaussian distribution. Consider the following didactic example where $B_Q = 1$ (the block prediction size).

Since -AND feeds earlier samples back into the model for making predictions, the likelihood of the target datapoints: $\{(x_i, y_i)\}_{i=1}^M$ for our -AND model is computed as follows:

$$\log p_{AND}(y_{1:M}|x_{1:M}, D_{context}) = \log \prod_{i=1}^M p(y_i|x_{1:i-1}, y_{1:i-1}, x_i, D_{context})$$

$$= \sum_{i=1}^M \log p(y_i|x_{1:i-1}, y_{1:i-1}, x_i, D_{context})$$

In contrast, consider the likelihood of -ND: $\log p_{ND}(y_{1:M}|x_{1:M}, D_{context})$. By Boole's Inequality (or Union Bound), we have that

$$\log p_{ND}(y_{1:M}|x_{1:M}, D_{context}) \leq \sum_{i=1}^M \log p(y_i|x_{1:M}, D_{context}) = \sum_{i=1}^M \log p(y_i|x_i, D_{context})$$

$(x_{1:i-1}, y_{1:i-1})$ provides relevant information for predicting the value of the function at $x_i$, e.g., nearby pixel values in image completion. As a result, it is likely the case that:

$$p(y_i|x_i, D_{context}) \leq p(y_i|x_{1:i-1}, y_{1:i-1}, x_i, D_{context})$$

Summing from $i = 1 \ldots M$, this means:

$$\log p_{ND}(y_{1:M}|x_{1:M}, D_{context}) \leq \log p_{AND}(y_{1:M}|x_{1:M}, D_{context})$$

As such, it is expected that the autoregressive not-diagonal variant's likelihood is higher than that of the non-autoregressive baselines.

## C.3 COMPARISON OF NPs WITH OTHER EXISTING METHODS FOR UNCERTAINTY ESTIMATION

Other popular methods which can perform uncertainty estimation, include and are not limited to MC-Dropout, Ensembles, Gaussian Processes (GPs), and Bayesian Neural Networks (BNNs).

Ensembles is an approximate Bayesian method which trains a group of neural networks on the same set of datapoints. The predictions of this group of neural networks are used to provide uncertainty predictions. Ensembles require retraining several models with gradient descent when new datapoints are received which is very costly.

GPs specify a Gaussian distribution over the function values that fit the datapoints. However, GPs scale cubically with the number of datapoints, making it only practical in settings with a small number of datapoints.

Bayesian Neural Networks is a stochastic neural network with a prior over weights trained using Bayesian inference. BNNs suffer their own respective challenges such as difficulty in tuning, difficulty in specifying weight priors, and cold posteriors. They also often perform worse compared to approximate bayesian methods.

# D    APPENDIX: IMPLEMENTATION, HYPERPARAMETER DETAILS, AND COMPUTE

## D.1    IMPLEMENTATION AND HYPERPARAMETER DETAILS

We use the implementation of the baselines from the official repository of TNPs (https://github.com/tung-nd/TNP-pytorch) and LBANPs (https://github.com/BorealisAI/latent-bottlenecked-anp). The datasets are standard for Neural Processes and are available in the same link. We follow closely the hyperparameters of TNPs and LBANPs. In CMANP, the number of blocks for the conditioning phase is equivalent to the number of blocks in the conditioning phase of LBANP. Similarly, the number of cross-attention blocks for the querying phase is equivalent to that of LBANP. We used an ADAM optimizer with a standard learning rate of $5e - 4$. We performed a grid search over the weight decay term $\{0.0, 0.00001, 0.0001, 0.001\}$. Consistent with prior work (Feng et al., 2023) who set their number of latents $L = 128$, we also set the number of latents to the same fixed value $L_I = L_B = 128$ without tuning. Due to CMANPs and CMABs architecture, they allow for varying embedding sizes for the learned latent values ($L_I$ and $L_B$). For simplicity, we set the embedding sizes to $64$ consistent with prior works (Nguyen & Grover, 2022; Feng et al., 2023). The block size for CMANP-AND is set as $B_Q = 5$. During training, CelebA (128x128), (64x64), and (32x32) used a mini-batch size of 25, 50, and 100 respectively. All experiments are run with 5 seeds. For the Autoregressive Not-Diagonal experiments, we follow TNP-ND and LBANP-ND (Nguyen & Grover, 2022; Feng et al., 2023) and use cholesky decomposition for our LBANP-AND experiments. Focusing on the efficiency aspect, we follow LBANPs in the experiments and consider the conditional variant of NPs, optimizing the log-likelihood directly.

## D.2    COMPUTE

All experiments were run on a Nvidia GTX 1080 Ti (12 GB) or Nvidia Tesla P100 (16 GB) GPU. 1-D regression experiments took 4 hours to train. EMNIST took 2 hours to train. CelebA (32x32) took 16 hours to train. CelebA (64x64) took 2 days to train. CelebA (128x128) took 3 days to train.

## D.3    COMPARISON OF MODEL PARAMETERS

Each CMAB consists of 2 self-attention blocks and 2 cross-attention blocks compared to LBANP's attention block which consists of 1 self-attention block and 1 cross-attention block. In our experiments, the models have 6 encoder layers (e.g., 6 CMABs) and 6 querying decoder layers (i.e., CrossAttention). As a result, CMANP uses an overall 30 attention blocks and LBANP uses an overall 18 attention blocks, i.e., CMANP uses approximately $67\%$ more parameters than LBANPs. Although CMANPs use more parameters than LBANPs, CMANPs ultimately use less memory (only constant!) since the number of inputs is the bottleneck in terms of memory usage for attention-based methods.

## D.4    RUNTIME

Previously, we analyzed the runtime for our method. Unfortunately, comparing the runtime of existing baselines is difficult as they have been optimized differently, making it hard to compare the runtimes fairly. NP models such as LBANPs and CMANPs have an architecture which interleaves modules, allowing for different modules to be computed in parallel at the same time for improved efficiency. For example, CMANPs compute encodings of the context dataset via: $L_i = \text{CrossAttention}(\text{SelfAttention}(L_{i-1}, \mathcal{D}_C)$ and retrieves information from this context dataset for prediction via: $X_{query}^i = \text{CrossAttention}(X_{query}^{i-1}, L_i)$. In an optimized codebase, computing $L_{i+1}$ and $X_i^{query}$ can actually be computed in parallel, resulting in a significantly more efficient model in terms of runtime. However, the publicly available codebase for LBANPs does not support this. Another example is that of Conditional Neural Processes (CNPs), a variant of NPs which leverages DeepSets. CNPs are able to efficiently compute updates via a rolling averaging mechanism. However, the available codebases do not support this by default either. Specialized implementations for comparing the runtime of NP methods are outside the scope of our work. Nonetheless, we detail below how to implement an efficient version of CMANPs.

### D.5 EFFICIENT IMPLEMENTATION

In our code, we implemented a sequential variant of CMANPs that computes each CrossAttention and Self-Attention module sequentially. However, computing parts of the stacked CMAB blocks in a model can be done in parallel to improve the processing speed. All CMAB blocks can compute the following costly operation $L'_B = \text{SelfAttention}(\text{CrossAttention}(L_B, \mathcal{D}_C))$ in parallel. In addition, CMAB can perform all updates to $L'_B$ in parallel as well. This is particularly important for the Autoregressive Not-Diagonal extension. When the prediction block size ($b_Q$) decreases, this corresponds to performing more CMAB updates since the predictions are made autoregressively. As such, a properly optimized codebase which computes in parallel would significantly reduce the runtime. Note that using the model this way would still be constant memory since the number of stacked CMAB blocks is a fixed hyperparameter.

