# OpenReview forum: "Memory Efficient Neural Processes via Constant Memory Attention Block"
_ICLR.cc/2024/Conference — Submitted to ICLR 2024_

### Official Review · Reviewer_PUEj · 2023-11-01

**Soundness:** 2 fair
**Presentation:** 3 good
**Contribution:** 3 good
**Rating:** 3
**Confidence:** 4

**Summary:**

In this paper, the authors address the crucial issue of memory efficiency in various applications, such as IoT devices, mobile robots, and other memory-constrained hardware environments. They highlight the problem of memory-intensive attention mechanisms, which can be a bottleneck in such settings. The authors introduce a novel solution through Constant Memory Attention Blocks (CMAB), which have the unique properties of being permutation invariant, performing computations in constant memory, and requiring updates in constant computation. This innovation allows CMABs to naturally scale to handle large amounts of inputs efficiently.

Building upon CMABs, the authors propose Constant Memory Attentive Neural Processes (CMANPs), which are not only scalable in the number of data points but also enable efficient updates. Additionally, they introduce an Autoregressive Not-Diagonal extension called CMANP-AND, which only requires constant memory, in contrast to the quadratic memory requirements of previous extensions. Experimental results demonstrate that CMANPs achieve state-of-the-art performance in tasks like meta-regression and image completion. Notably, CMANPs excel in memory efficiency by requiring constant memory, making them significantly more efficient than prior state-of-the-art methods.

This paper presents an innovative solution to the challenge of memory efficiency in memory-constrained hardware environments. CMANPs, built on CMABs, offer scalable and memory-efficient attention mechanisms, improving the efficiency and effectiveness of various applications.

**Strengths:**

The topic is interesting and memory efficiency is very important.
The proposed CAMB is designed to be updated by an iterative process.
The explanations of the method are clear and there are sufficient baselines as comparisons.

**Weaknesses:**

The iterative process of the CMANP will be potentially inefficient.
The experiments show the improvements are not significant enough since the improvements are small.
The experiments only have two datasets to evaluate which is not enough.

**Questions:**

Is the method useful for more datasets?

---

> ### Author Response · Authors · 2023-11-19
> **Response to Reviewer PUEj**
>
> We would like to thank the reviewer for the comments and feedback.
>
> > The iterative process of the CMANP will be potentially inefficient.
>
> We would like to clarify that the efficiency of the iterative process is controlled by the user by setting the value of the constant $b_C$ (i.e., the size in which to chunk the input data).
>
> If a large value of $b_C$ is selected, CMANPs will only require a single iteration, making it similarly as efficient as LBANPs (Feng et al., 2023), i.e., an efficient variant of Neural Processes.
> If a small value of $b_C$ is selected, CMANPs will only require a small constant amount of memory at the expense of needing more iterations (i.e., time). Thus, the method provides a way to control the tradeoff between memory and computational time.
>
> We address this concern further by detailing in the Appendix: Section D.5 shows how parts of the iterative process can be parallelized across CMAB blocks for further efficiency.
>
> ----
>
> Feng, Leo, Hossein Hajimirsadeghi, Yoshua Bengio, and Mohamed Osama Ahmed. "Latent Bottlenecked Attentive Neural Processes." In The Eleventh International Conference on Learning Representations. 2023.
>
> > The experiments show the improvements are not significant enough since the improvements are small.
>
> In the title and introduction, we have tried to emphasize that the focus is on improving the memory efficiency in Neural Processes while maintaining performance competitive with state-of-the-art. As such, the main focus of comparison is on the memory usage (Figure 3) rather than the log-likelihood (Table results).
>
> Figure 3 shows that CMANPs use far less memory than prior state-of-the-art methods by leveraging our Constant Memory Attention Block. More specifically, Figure 3 (left) shows empirically that CMANP (CMAB-based model) only requires constant memory regardless of the number of context data points making it significantly more efficient than models based on Perceiver's iterative attention (LBANPs) and Transformers (TNPs) which scale linearly and quadratically respectively. Figure 3 (right) also shows that CMANP-AND is significantly more memory efficient than TNP-ND and LBANP-ND which scale quadratically.
>
> In terms of log-likelihood, our table results show that CMANPs achieve similar performance to state-of-the-art while being able to scale to more datapoints. For instance, we showed that CMANP-AND can be trained on CelebA (64x64) and CelebA (128x28) while prior Not-Diagonal variants were not able to be trained due to requiring a quadratic amount of memory.
> Notably, on (CelebA 32x32), CMANP-AND achieved $6.31 \pm 0.04$ log-likelihood (higher is better) significantly outperforming both TNP-ND and LBANP-ND achieved $5.48 \pm 0.02$ and $5.57 \pm 0.03$ respectively.
>
>
>
> > The experiments only have two datasets to evaluate which is not enough.
>
> We would like to clarify that we have evaluated on a wide range of tasks. Specifically, we have evaluated on 6 popular tasks. However, due to space limitations and the paper focusing on Neural Processes, several tasks are included in the Appendix.
>
> In this work, we evaluated on 4 common Neural Processes problem settings: CelebA (32x32, 64x64, 128x128), EMNIST, GP Regression Tasks (RBF Kernel and Matern Kernel), and Contextual Bandits (Results in Appendix: Section B.2).
>
> Going beyond Neural Processes, in Appendix: Section B.1, we showcased the efficacy of the Constant Memory Attention Block on event prediction (Temporal Point Processes) by replacing the transformer in Transformer Hawkes Process (a popular method in TPP literature) with our proposed Constant Memory Attention Block, showing competitive results on two popular datasets in the TPP literature: Mooc and Reddit.
>
> > Questions:
> > Is the method useful for more datasets?
>
> Please see the above comment where we detail the diverse range of tasks which we evaluated on.

---

> ### Author Response · Authors · 2023-11-22
> **Message to Reviewer PUEj**
>
> Dear Reviewer PUEj,
>
> We appreciate your time and consideration. We have addressed all your concerns in the comments below. We have (1) clarified that the efficiency of the iterative process is controlled by the user, (2) shown that the improvements in terms of memory are significant, and (3) clarified that we have evaluated on $6$ tasks in total.
>
> Could you please let us know if you have any further concerns? We are happy to address any further concerns you have. Any feedback would be highly appreciated. We look forward to hearing from you.

---

### Official Review · Reviewer_SwB8 · 2023-11-02

**Soundness:** 3 good
**Presentation:** 4 excellent
**Contribution:** 3 good
**Rating:** 6
**Confidence:** 3

**Summary:**

The authors present a new approach to latent variable models based on Neural Processes that use a more efficient attention mechanism that has favorable properties from a memory perspective.

**Strengths:**

- The proposed approach is novel as applied to Neural Processes, and the update mechanism is unique insofar as it is computationally efficient. That adding new data involves a constant-time update allows the approach to scale very efficiently.
- The permutation invariance of the attention mechanism is well-exploited in this setting; cross-attention is equally a good approach for efficiency.
- The paper is well-written and relatively concise.
- The results as shown are compelling, demonstrating god improvements on prior NP methods, including those leveraging attention.

**Weaknesses:**

- The authors might be more explicit about the ways in which memory is a meaningful limitation in conventional Neural Process methods. It may be possible to more-explicitly show both memory and method performance in a plot or table to illustrate the interplay.
- While the authors demonstrate the relationship between block size and end-to-end time cost (in the appendix), doing so for their proposed approach compared to other latent variable modeling approaches might be helpful to contextualize CMANPs in terms of not only memory, but also computation time.

**Questions:**

- The authors may care to briefly explain here and in the manuscript: why is permutation invariance important for latent variable/neural process modeling?
- Table 3 might be improved by including "ours" in relevant rows for readability.
- The equation at the end of section 3 describing CMAB-AND is rather dense/difficult to read, even given an understanding of each of the components. An explanation of the update mechanism aside from the equation might help clarify.
- As described in section 4.1, how much does increasing the number of iterations affect the performance of the model?
- What were the resources used for training, and what qualified as "too computationally expensive" for Non-Diagonal variants trained on CelebA?

---

> ### Author Response · Authors · 2023-11-19
> **Response to Reviewer SwB8**
>
> We would like to thank the reviewer for their very constructive feedback and their support.
>
> > The authors might be more explicit about the ways in which memory is a meaningful limitation in conventional Neural Process methods.  It may be possible to more-explicitly show both memory and method performance in a plot or table to illustrate the interplay.
>
>
> Thank you for your suggestion! We have updated the introduction to make more explicit how memory is a limitation in Neural Processes. The memory gains and method performance can be inferred from Figure 3 and the table results. If it would help, we can prepare the plot for the camera-ready.
>
> > While the authors demonstrate the relationship between block size and end-to-end time cost (in the appendix), doing so for their proposed approach compared to other latent variable modeling approaches might be helpful to contextualize CMANPs in terms of not only memory, but also computation time.
>
> We have included plots in the Appendix (Figure 11) which provide several comparisons of the computation time (i.e., runtimes).
>
> In Figure 11a, we compare the runtime of the update process of CMAB-based NP (CMANPs) with Perceiver's iterative attention-based NP model (LBANPs) and a transformer-based NP model. We see that the CMAB-based model only requires a constant amount of time to perform the update. In contrast, Perceiver's iterative attention-based model's update runtime scales linearly and Transformer model's update runtime scales quadratically.
>
> In Figure 11b, we compare the runtime of querying (performing inference) of CMANP with LBANP (Perceiver's iterative attention-based model), TNP (Transformer-based model). We see that CMANP and LBANP stay constant due to only needing a fixed number of latents to perform inference while the transformer-based model (TNP) scales quadratically in runtime.
>
> > Questions:
> > The authors may care to briefly explain here and in the manuscript: why is permutation invariance important for latent variable/neural process modeling?
>
>
> From a theoretical and historical perspective, Neural Processes (Garnelo et al., 2018a;b) were originally inspired by Gaussian Processes which are permutation invariant in the context.
>
> However, from a practical perspective, permutation invariance is a generally useful property for versatile models. For example, self-attention is by default permutation-invariant. As a result, Transformers have become widely applicable by augmenting the input with various modality-specific components such as a positional encoding for language modelling or patches of an image for computer vision. Similarly, NPs have been applied to temporal data and images, leveraging modality-specific components. We refer the reviewer to the survey by Jha et al., 2023 for a broad overview of how NPs have leveraged modality-specific components for different tasks.
>
>
> ---
> Garnelo, Marta, Dan Rosenbaum, Christopher Maddison, Tiago Ramalho, David Saxton, Murray Shanahan, Yee Whye Teh, Danilo Rezende, and SM Ali Eslami. "Conditional neural processes." In International conference on machine learning, pp. 1704-1713. PMLR, 2018.
>
> Garnelo, Marta, Jonathan Schwarz, Dan Rosenbaum, Fabio Viola, Danilo J. Rezende, S. M. Eslami, and Yee Whye Teh. "Neural processes." arXiv preprint arXiv:1807.01622, 2018.
>
> Saurav Jha, Dong Gong, Xuesong Wang, Richard E. Turner, Lina Yao. "The neural process family: Survey, applications and perspectives." arXiv preprint arXiv:2209.00517, 2023.
>
> > Table 3 might be improved by including "ours" in relevant rows for readability.
>
> Thank you for your suggestion. We have updated Table 3 to include "ours" in the relevant rows to improve readability.
>
> > The equation at the end of section 3 describing CMAB-AND is rather dense/difficult to read, even given an understanding of each of the components. An explanation of the update mechanism aside from the equation might help clarify.
>
> Thank you for letting us know! We have updated Section 3 to improve the clarity, including an explanation of the update mechanism.
>
> > As described in section 4.1, how much does increasing the number of iterations affect the performance of the model?
>
> Could you please clarify which iterations you are referring to?
>
> > What were the resources used for training, and what qualified as "too computationally expensive" for Non-Diagonal variants trained on CelebA?
>
> In the experiment, we used an Nvidia P100 (16GB) GPU. "Too computationally expensive" in this case refers to experiments where the model could not be trained due to memory usage.

---

### Official Review · Reviewer_GqA3 · 2023-11-02

**Soundness:** 3 good
**Presentation:** 2 fair
**Contribution:** 1 poor
**Rating:** 3
**Confidence:** 3

**Summary:**

The author proposed the constant memory attention block to make memory usage efficient in attentional neural processes.

**Strengths:**

The overall workflow is clear and the authors provide enough details on the proposed approach. The authors incorporated a rich set of baselines in the comparison.

**Weaknesses:**

At the very core of the algorithm, section 3.1.2, it seems the proposed approach basically achieves the constant memory by splitting the batch of input data into numerous slices of constant size, but at the same time, more operations are involved in the form of "CA(LB,D) = UPDATE(D1,UPDATE(D2, . . .UPDATE(...". Analysis is needed on if the added computational and operational complexity is worth the saved memory space. For such splitting of input batches, it seems to be more of an engineering design choice to be done by low level programming of GPU drive instead of tackling it at the high level architecture design.

It is always easy to modify some architectures/operations from current state of the art network structures for it to have approximately the same performance on a specific type of (meta learning) task. In my perspective, to claim concrete contribution of advancing current state of the arts based on network architecture engineering, the authors need to perform experiments on a wider range of tasks to show the versatility of proposed approach. The work seems pretty incremental on top of state of the arts attention models.

**Questions:**

is CMAB mechanism applicable to other types of learning models in addition to neural processes? Clarification is needed.

---

> ### Author Response · Authors · 2023-11-19
> **Response to Reviewer GqA3  (1/2)**
>
> We would like to thank the reviewer for the comments and feedback.
>
> > At the very core of the algorithm, section 3.1.2, it seems the proposed approach basically achieves the constant memory by splitting the batch of input data into numerous slices of constant size, but at the same time, more operations are involved in the form of "CA(LB,D) = UPDATE(D1,UPDATE(D2, . . .UPDATE(...". Analysis is needed on if the added computational and operational complexity is worth the saved memory space. For such splitting of input batches, it seems to be more of an engineering design choice to be done by low level programming of GPU drive instead of tackling it at the high level architecture design.
>
> As you have described, CMAB achieves constant memory by leveraging its efficient updates property and applying it repeatedly.  However, CMAB's efficient updates property is not achievable by just low-level programming of a GPU. In Section 3.1.1, we provided proof of CMAB's constant computation updates. The proof is reliant on a special set of learnable latents $L_B$ within our proposed Constant Memory Attention Block. As such, the efficient updates property is specific to CMAB's high-level architectural design and is not shared with other attention methods such as Perceiver's iterative attention or Transformer. We can show that CMAB's efficient updates property is not shared by Perceiver's iterative attention.
>
> Furthermore, a naive implementation of CMAB's update operation is not stable and can easily run into overflow issues. As such, we included in Appendix A.1, a non-trivial derivation of a stable implementation which avoids these issues.
>
> Beyond proving the constant memory complexity, CMAB's unique ability to perform constant computation updates is also highly useful. Notably, since the updates are constant computation and do not depend on the number of prior tokens, CMAB is very useful in streaming data settings and also scales very well in contrast to prior attention methods which scale linearly or quadratically. Finally, CMAB's update operation does not require storing the previous tokens.
>
> **To the best of our knowledge, no prior attention methods achieve CMAB's $3$ important properties together:** (1) performs updates in constant computation, (2) computes its output in constant memory, and (3) is permutation invariant.
>
> ---
>
> Here, we describe why Perceiver's iterative attention does not share CMAB's efficient updates property.
>
> Perceiver's Iterative Attention when stacked works as follows: $L_{i+1} = \mathrm{SelfAttention(CrossAttention(}L_{i}, \mathcal{D}))$
> where $\mathrm{SA}$ refers to Self-Attention, $\mathrm{CA}$ refers to CrossAttention, and $L_i$ and $L_{i+1}$ are the latent inputs and outputs for the $i$-th block respectively.
>
> In our work, we proved that the output of CrossAttention can be updated in constant computation per datapoint via a rolling summation given that the query vectors are constants. In this case, $L_0$ is a learned constant. However, when new datapoints $\mathcal{D}_U$ are added to the set of input tokens, i.e., $\mathcal{D} \cup \mathcal{D}_U$,
>
> the input latents for later attention modules ($L_{i}$ where $i > 0$) change and is thus not a constant, so Perceiver's iterative attention does not have the constant computation updates property and would always require re-computing with all the input tokens from scratch.
>
> CMABs are different by first computing $L_B' = \mathrm{SelfAttention(CrossAttention(}L_B, \mathcal{D}))$ and then $L_{i+1} = \mathrm{SelfAttention(CrossAttention(}L_i , L_B))$. Notably, $L_B$ is a learned constant within the block, so computing the updated $L_B'$ given the new data can be done in constant computation. Furthermore, since $L_i$ and $L_B$ have constant size, thus computing $L_{i+1}$ can be done in constant computation, resulting in CMABs' constant computation updates property.

---

> ### Author Response · Authors · 2023-11-19
> **Response to Reviewer GqA3 (2/2)**
>
> > In my perspective, to claim concrete contribution of advancing current state of the arts based on network architecture engineering, the authors need to perform experiments on a wider range of tasks to show the versatility of proposed approach.
>
> We would like to clarify that we have evaluated on a wide range of tasks. Specifically, we have evaluated on 6 popular tasks. However, due to space limitations and the paper focusing on Neural Processes, several tasks are included in the Appendix.
>
> In this work, we evaluated on 4 common Neural Processes problem settings: CelebA (32x32, 64x64, 128x128), EMNIST, GP Regression Tasks (RBF Kernel and Matern Kernel), and Contextual Bandits (Results in Appendix: Section B.2).
>
> Going beyond Neural Processes, in Appendix: Section B.1, we showcased the efficacy of the Constant Memory Attention Block on next-event prediction (Temporal Point Processes) by replacing the transformer in Transformer Hawkes Process (a popular method in TPP literature) with our proposed Constant Memory Attention Block, showing competitive results on two popular datasets in the TPP literature: Mooc and Reddit.
>
> > Questions:
> > is CMAB mechanism applicable to other types of learning models in addition to neural processes? Clarification is needed.
>
> Yes -- CMAB is applicable to other types of learning models. Going beyond Neural Processes, we also showed the efficacy of CMAB on next-event prediction (Temporal Point Processes) models. However, due to space limitations, these details were included in the Appendix (Appendix B.1).
>
> We have updated the camera-ready to clarify that indeed CMAB applies to other learning models and point out the additional results in the Appendix.

---

> ### Author Response · Authors · 2023-11-22
> **Message to Reviewer GqA3**
>
> Dear Reviewer GqA3,
>
> We appreciate your time and consideration. We have addressed all your concerns in the comments below. We have (1) clarified that the properties of CMABs are (i) not achievable by low-level programming of the GPU and (ii) not shared by prior methods, (2) detailed that we have evaluated on a wide range of tasks, including image completion, GP regression, and contextual bandits, and (3) explained that the CMAB mechanism is applicable to other types of learning models; however, due to space limitations, these details are included in the appendix.
>
> Could you please let us know if you have any further concerns? We are happy to address any further concerns you have. Any feedback would be highly appreciated. We look forward to hearing from you.

---

### Official Review · Reviewer_wchU · 2023-11-09

**Soundness:** 3 good
**Presentation:** 3 good
**Contribution:** 3 good
**Rating:** 8
**Confidence:** 4

**Summary:**

The paper presents the Constant Memory Attention Block (CMAB), a new attention mechanism that facilitates memory-efficient operations, crucial for applications where computational resources are limited. This approach streamlines the modeling process without the high memory demand typical of advanced attention mechanisms.

Expanding on CMAB, the authors propose the Constant Memory Attentive Neural Processes (CMANPs) along with an autoregressive variant, CMANP-AND. This final model iteration maintains the constant memory efficiency characteristic of CMANPs, contrasting with the less scalable Not-Diagonal methods that require exponentially more memory as the number of target data points increases. CMANP-AND's design is essential for efficiently processing a full covariance matrix, a memory-hungry task, thus enabling it to achieve leading performance on Neural Process benchmarks with greater memory efficiency.

**Strengths:**

- **Clarity**: The paper is well-written and structured, offering a clear explanation of its contributions and comparisons with existing Neural Processes (NP) models, and provides results in a manner that is easy to follow.
- **Quality**: The model is evaluated against established NP benchmarks. The constant memory advantage is supported by theoretical proofs and empirical evidence,
- **Originality**: The paper's CMAB proposes an innovative approach to attention mechanisms, employing constant memory to enhance NP, which represents a notable advancement over the memory constraints associated with conventional models.
- **Significance**: The CMABNP-AND's ability to bring high-performing deep learning to low-resource settings by addressing and overcoming the scalability limitations of contemporary models is a significant stride in widening their applicability.
- **Applicability**: The paper is particularly notable for its application potential, demonstrating how the model can be scaled to tackle harder problems, such as higher-resolution challenges, a common limitation in existing models.

**Weaknesses:**

1. The main weakness that I see in this work lies in the trade-off between constant memory usage and computational efficiency. To maintain constant memory, the input data is divided into batches, requiring multiple iterative updates rather than a single computation. This approach, while memory-efficient, could potentially lead to increased computational time, as the number of updates is inversely proportional to the batch size. The figures provided offer valuable insights into the relationship between block size and time within the CMABNP model itself. However, to fully substantiate the efficiency claims, it would be advantageous to expand the experimental setup to include comparisons of the CMABNP model with traditional update rules. Such an experiment could involve running the same model with and without the new update rule e.g. across CalebA32, 64, and 128. This would not only highlight the benefits of the proposed method in a more diverse context but also guide users in understanding the practical trade-offs when implementing this model in real-world scenarios.
2. While the paper posits the CMABNP model as particularly useful for streaming data settings like e.g. contextual bandits and Bayesian optimization, it lacks empirical evidence to support these claims. Given the relevance of such applications, it would strengthen the paper to include experiments that demonstrate the model's performance in these specific scenarios. Furthermore, a direct comparison with current state-of-the-art implementations, such as PFNs4BO [1], would not only validate the model's utility in practical settings but also offer a clearer understanding of its position relative to existing methods.
3. The paper's discussion of implementing CMAB blocks in parallel for efficiency raises questions about the actual execution framework used in their experiments, particularly since the model is intended for low-resource settings where parallel processing might not be an option. It is not entirely clear to me whether the efficient parallel implementation was employed in the experiments or not. To address this, it would be constructive to see a comparison of the CMAB model's performance with parallel processing enabled versus a purely sequential implementation. Such an analysis would not only clarify the model's operational requirements but also help in understanding the practicality and scalability of CMAB in various computational environments.
4. Lastly, the paper could be enhanced in the provision of implementation code.

[1] PFNs4BO: In-Context Learning for Bayesian Optimization, Samuel Müller and Matthias Feurer and Noah Hollmann and Frank Hutter, ICML 2023

**Questions:**

1. Could you detail the computational efficiency of the update process and provide comparative runtimes for the models?
2. Is there empirical evidence to support the model's utility in streaming data settings?
3. Can you clarify whether parallel computation was used in the experiments? And if so, how does the model perform when processed sequentially, as would be common in low-resource environments?
4. Could you share the code?
5. Could you provide in Figure 8, in the Appendix, visualizations that would clearly show out-of-distribution prediction?

---

> ### Author Response · Authors · 2023-11-19
> **Response to Reviewer wchU**
>
> We would like to thank the reviewer for the detailed comments and helpful feedback. We are pleased to see your enthusiasm for our work.
>
> > Questions:
> > Could you detail the computational efficiency of the update process and provide comparative runtimes for the models?
>
> **Computational Efficiency of the update process:** Given the original input data $\mathcal{D}$ and a new set of datapoints $\mathcal{D}_{U}$ to update the model with, the original update process involves computing CMAB from scratch using $\mathcal{D} \cup \mathcal{D}_U$. In contrast, CMAB's update process leverages a rolling summation trick using only $\mathcal{D}_U$ to reduce the overall amount of computation.
>
> As a result, the amount of computation required for CMAB when using the original update process is $\mathcal{O}((|\mathcal{D}| + |\mathcal{D}_U|) |L_B| + |L_B|^2 + |L_B| |L_I| + |L_I|^2)$.
>
> In contrast, CMAB's update process requires $\mathcal{O}(|\mathcal{D}_U| |L_B| + |L_B|^2 + |L_B| |L_I| + |L_I|^2)$ computation. Since $|L_I|$ and $|L_B|$ are pre-specified constants, this update process is constant computation per new datapoint.
>
> The computational efficiency gain by leveraging CMAB's update process is thus $\mathcal{O}(|\mathcal{D})| |L_B|)$, meaning that the efficiency gain scales with the size of the input data. This makes CMABs naturally extra useful in streaming data settings where the amount of contextual data increases over time.
>
> **Runtime Comparison:** We have included plots in the Appendix (Figure 11) which provide several comparisons of the runtimes.
>
> In Figure 11a, we compare CMANP using the efficient update process and the traditional update process, showing that the efficient update process is initially similar in runtime to the traditional update process. However, as the number of context datapoints increases (i.e., updates are performed) over time, the traditional update process' runtime grows linearly while our proposed efficient update process' runtime remains constant regardless of the number of tokens.
>
> In Figure 11a, we also compare the runtime of the update process of CMAB-based NP (CMANPs) with Perceiver's iterative attention-based NP model (LBANPs) and a transformer-based NP model. We see that the CMAB-based model only requires a constant amount of time to perform the update. In contrast, Perceiver's iterative attention-based model's update runtime scales linearly and Transformer's update runtime scales quadratically.
>
>
> In Figure 11b, we compare the querying (inference) runtime of CMANP with LBANPs (Perceiver's iterative attention-based model), TNPs (Transformer-based model). We see that CMANPs and LBANPs stay constant while the transformer-based model (TNP) scales quadratically in runtime.
>
>
> > Is there empirical evidence to support the model's utility in streaming data settings?
>
> A main point about streaming data setting is that the data is received batch by batch. As such, CMAB and CMANP are highly useful as they guarantee that the updates will be performed in constant computation and constant memory. We showed empirically their constant computation updates property in Figure 11 (Appendix) and constant memory property in Figure 3. In contrast, alternative attention methods would require computing from scratch, resulting in computation scaling linearly or quadratically. Furthermore, CMAB and CMANP do not require storing the previous context data when performing their updates, making it highly valuable in low-resource settings. Finally, in the appendix, we also included results of CMANPs applied to Contextual Bandits (a streaming data setting), showing CMANPs are competitive with prior state-of-the-art NPs. As such, we believe that a future application of CMABs (not limited to Neural Processes) to various streaming data settings would be an interesting research direction.
>
> > Can you clarify whether parallel computation was used in the experiments? And if so, how does the model perform when processed sequentially, as would be common in low-resource environments?
>
> Our implementation is a sequential version. We detailed the parallel computation in the Appendix as potentially useful information for future implementations. We have clarified this in the updated draft.
>
> > Could you share the code?
>
> We will share the code alongside the camera-ready version. If it would help with the review process, we can prepare an anonymized version.
>
> > Could you provide in Figure 8, in the Appendix, visualizations that would clearly show out-of-distribution prediction?
>
> As requested, we have added plots (Figure 9) to showcase CMANPs efficacy on out-of-distribution predictions. In the plots, the context datapoints are only sampled from (left plot) $[-1.0, 2.0]$, (middle plot) $[-2.0, 1.0]$, and (right plot) $[-2.0, -1.0] \cup [1.0, 2.0]$ while the uncertainty estimates are shown for $[-2.0, 2.0]$.

---

> > ### Comment · Reviewer_wchU · 2023-11-22
> >
> > Thank you very much for the detailed response and new content! I will increase the Contribution score.

---

### Meta-Review · Area_Chair_t2d4 · 2023-12-03

**Metareview:**

The paper proposes a scalable variant of attentive Neural Processes, by introducing a constant memory attention block.

One reviewer (score 8) was supportive of the work, another reviewer was moderately supportive (score 6), and two reviewers were critical of the work (scores 3, and 3).

Overall, the main expressed criticism is the lack of novelty (incrementality of the idea) and the insufficient transparency in analyzing the trade-off between the memory efficiency of the proposed method and its computational complexity. Furthermore, it is not well-demonstrated to what extent the proposed method solves practical memory-limited tasks, which are not solvable by prior practices. I agree that the expressed limitations need to be better addressed in the manuscript.

In addition, I believe the authors ambiguously position the novelty of the paper as both (i) a general-purpose memory-efficient attention block, and (ii) a more scalable Neural Process. While the tone is on (i) the experiments focus on (ii). To validate (i) you need to integrate your method into a broader family of general-purpose transformer architectures and show the impact on memory efficiency outside the realm of Neural Processes.

The AC recommends rejection of the paper in its current state and advises the reviewers to consider the provided suggestions for the next submission.

**Justification For Why Not Higher Score:**

Limited analysis of the trade-off between the proposed memory efficiency to the computational complexity.

**Justification For Why Not Lower Score:**

N/A

---

### Decision · Program_Chairs · 2024-01-16

Reject